

# Differences in microphysical properties of cirrus at high and mid-latitudes

Elena De La Torre Castro[1, 2, 3], Tina Jurkat-Witschas[1], Armin Afchine[4], Volker Grewe[1, 3], Valerian Hahn[1, 2], Simon Kirschler[1, 2], Martina Krämer[4, 2], Johannes Lucke[1, 3], Nicole Spelten[4], Heini Wernli[5], Martin Zöger[6], and Christiane Voigt[1, 2]

[1]Institute of Atmospheric Physics, German Aerospace Center, DLR, Oberpfaffenhofen, Germany
[2]Institute of Atmospheric Physics, Johannes Gutenberg University Mainz, Mainz, Germany
[3]Faculty of Aerospace Engineering, Delft University of Technology, Delft, the Netherlands
[4]Institute for Energy and Climate Research, Research Center Jülich, Jülich, Germany
[5]Institute for Atmospheric and Climate Science, ETH Zurich, Zurich, Switzerland
[6]Institute for Flight Experiments, German Aerospace Center, DLR, Oberpfaffenhofen, Germany
**Correspondence:** Elena De La Torre Castro (Elena.delaTorreCastro@dlr.de)

**Abstract.** Despite their proven significance for the atmospheric radiative energy budget, the effect of cirrus on climate and the magnitude of their modification by human activity is not well quantified. Besides anthropogenic pollution sources on the ground, aviation has a large local effect on cirrus microphysical and radiative properties via the formation of contrails and their transition to contrail cirrus. To investigate the anthropogenic influence on natural cirrus, we compare the microphysical

properties of cirrus measured at mid-latitude regions (ML, $< 60°$ N) that are often affected by aviation and pollution with cirrus measured in the same season in comparatively pristine high latitudes (HL, $\geq 60°$ N). The number concentration, effective diameter, and ice water content of the observed cirrus are derived from in situ measurements covering ice crystal sizes between between 2 and 6400 µm collected during the CIRRUS-HL campaign (CIRRUS in High Latitudes) in June and July 2021. We analyse the dependence of cirrus microphysical properties on altitude and latitude and demonstrate that the median ice

number concentration is by an order of magnitude larger in the measured mid-latitude cirrus with $0.0086$ cm$^{-3}$ compared to $0.001$ cm$^{-3}$ in high-latitude cirrus. Ice crystals in mid-latitude cirrus are on average smaller than in high-latitude cirrus, with a median effective diameter of 165 µm compared to 210 µm and the median ice water content in mid-latitude cirrus is higher ($0.0033$ g m$^{-3}$) than in high-latitude cirrus ($0.0019$ g m$^{-3}$). In order to investigate the cirrus properties in relation to the region of formation, we combine the airborne observations with 10-day backward trajectories to identify the location of

cirrus formation and the cirrus type: in situ or liquid origin cirrus, depending on whether there is only ice or also liquid water present in the cirrus history, respectively. The cirrus formed and measured at mid-latitudes (M-M) have particularly high ice number concentration and low effective diameter. This is very likely a signature of contrails and contrail cirrus, which is often observed in the in situ origin cirrus type. In contrast, the largest effective diameter and lowest number concentration were found in the cirrus formed and measured at high latitudes (H-H) along with the highest relative humidity over ice (RH$_i$). On

average, in-cloud RH$_i$ was above saturation in all cirrus. While most of the H-H cirrus were of in situ origin, the cirrus formed at mid-latitudes and measured at high latitudes (M-H) were mainly of liquid origin. A pristine Arctic background atmosphere with scarce availability of ice nuclei and the extended growth of few nucleated ice crystals may explain the observed RH$_i$



and size distributions. The M-H cirrus are a mixture of the properties of M-M and H-H cirrus (preserving some of the initial properties acquired at mid-latitudes and transforming under Arctic atmospheric conditions). Our analyses indicate that part of the cirrus found at high latitudes are actually formed at mid-latitudes, and therefore affected by mid-latitude air masses, which have a greater anthropogenic influence.

## 1 Introduction

Cirrus clouds are ice clouds at high altitudes ($> 8$ km at mid-latitudes and $> 12$ km in the tropics) and cover about one third of the Earth surface (Sassen et al., 2008). On the global average, cirrus and, in particular, thin cirrus exert a net warming effect on the atmosphere by absorbing thermal-infrared radiation from the Earth surface and re-emitting it back to the surface (Liou, 1986; Gasparini et al., 2018). In addition, cirrus reflect solar radiation. This implies that less radiation reaches the surface, which leads to a cooling effect (Hong and Liu, 2015; Gasparini et al., 2018; Marsing et al., 2023). In the absence of solar radiation (e.g. during nights and polar winter seasons) cirrus exert a greenhouse effect and warm the atmosphere (Heymsfield et al., 2017b; Gasparini et al., 2018; Marsing et al., 2023). In contrast to thin cirrus, thicker cirrus reflect more solar radiation back to space and thereby tend to have a net cooling effect on the atmosphere (Choi and Ho, 2006). In addition, regional and seasonal variations modulate the cirrus radiative forcing. On average, cirrus at high latitudes have a net warming effect throughout the year and cirrus at mid-latitudes tend to warm during the winter and cool during the summer (Hong and Liu, 2015).

The cirrus impact on the radiative energy budget depends on their macrophysical and microphysical properties: number concentration (N), effective diameter (ED), ice water content (IWC), and on the shape of the ice crystals (Liou, 1986; Wendisch et al., 2005, 2007). These properties display a large variability. The properties of cirrus are affected by thermodynamic (temperature and relative humidity) and dynamic conditions (updraft velocity), as well as on aerosol load and chemical composition including anthropogenic influences (Hendricks et al., 2011; Patnaude and Diao, 2020; Maciel et al., 2022). The combination of these factors determines the nucleation process and the number, size, and shape of the ice crystals in the cloud.

Cirrus form due to the lifting and the resulting cooling of air either by homogeneous nucleation of soluble aerosol particles or heterogeneous nucleation of insoluble ice nucleating particles (INPs) (Kärcher and Lohmann, 2002; Hoose and Möhler, 2012). Homogeneous nucleation occurs most frequently at strong cooling rates and high supersaturations with respect to ice ($RH_i \sim 150\%$). Gravity waves, for example, can trigger the nucleation of homogeneously formed ice crystals, which can reach concentrations from 1 to $10$ cm$^{-3}$ (Kärcher and Lohmann, 2002; Hendricks et al., 2011; Jensen et al., 2013). However, in the presence of INPs, heterogeneous nucleation takes place at lower $RH_i$ than homogeneous freezing. The competition between both nucleation mechanisms together with sedimentation processes and temperature fluctuations control the cirrus microstructure (Spichtinger and Gierens, 2009; Kärcher et al., 2022).

Another classification based on the Lagrangian origin of the air in which the cirrus form was applied in several studies (Krämer et al., 2016; Luebke et al., 2016; Wernli et al., 2016; Heymsfield et al., 2017b; Voigt et al., 2017; Krämer et al., 2020). In situ origin cirrus form most likely below $-38$ °C in strong updrafts by homogeneous nucleation (Kärcher and Lohmann,





2002) and potentially at higher temperatures in slow updrafts by heterogeneous nucleation (Kärcher and Lohmann, 2003). Liquid origin cirrus form from mixed-phase clouds at lower altitudes that glaciate as they ascend and rise to the cirrus heights. Usually, liquid origin cirrus have higher IWC than in situ origin cirrus (Krämer et al., 2016; Luebke et al., 2016).

Cirrus at mid-latitudes have been an important object of study over the last decade (Heymsfield et al., 2013; Krämer et al., 60 2016; Luebke et al., 2016; Voigt et al., 2017; Krämer et al., 2020) and measurements of Arctic cirrus have become an urgent topic to address in recent years (Wolf et al., 2018; Marsing et al., 2023). Such measurements are particularly relevant for understanding the regional and seasonal variations of Arctic cirrus and their relationship with the phenomenon of Arctic amplification (Schmale et al., 2021; Shupe et al., 2022; Wendisch et al., 2023). The rapid and continuous warming of the Arctic is expected to reduce the temperature gradient between mid- and high latitudes with disputed consequences on mid-latitude 65 extreme weather (Wendisch et al., 2017). Additionally, the possibility of geoengineering the Arctic through cirrus seeding to revert climate change has been discussed in the last decade (Storelvmo et al., 2014; Muri et al., 2014). Thus, understanding the processes affecting the microphysical properties of cirrus at high latitudes is essential for answering current and future urgent questions in climate research.

The number concentration of ice crystals is strongly dependent on the updraft and supersaturation driving the nucleation 70 process of ice crystals on different INP types (Kärcher and Lohmann, 2003). These parameters are mainly determined by the prevalent meteorological conditions (Muhlbauer et al., 2014). Balloon-borne measurements of ice crystals shapes and number concentrations of Arctic cirrus close to Kiruna (Sweden) were analysed by Wolf et al. (2018), who found lower concentrations compared to the measurements at mid-latitudes (Krämer et al., 2016; Luebke et al., 2016). Gayet et al. (2004) compared microphysical properties of cirrus in the Northern and Southern Hemisphere and found higher particle concentrations and lower 75 effective diameters in the more polluted Northern Hemisphere. However, it has not been confirmed by the global distributions of N from satellite retrievals of recent studies (Sourdeval et al., 2018; Krämer et al., 2020).

The ice water content of cirrus in the Arctic, mid-latitude and tropical regions varies over several orders of magnitude within the same temperature ranges (Schiller et al., 2008; Luebke et al., 2013; Heymsfield et al., 2017a; Krämer et al., 2020; Marsing et al., 2023). Studies with direct comparisons of N, ED, and IWC from dedicated measurements at high and mid-latitudes 80 are scarce but necessary to better understand the different processes driving cirrus formation. In addition, seasonal changes in cirrus and variability of dynamical processes in the troposphere complicate direct comparisons between data from different field campaigns.

Aircraft emissions can influence natural cloudiness not only through the injection of aerosols in the upper troposphere, but also through the direct formation of contrails (Voigt et al., 2010, 2011; Burkhardt and Kärcher, 2011; Tesche et al., 2016; 85 Marjani et al., 2022; Li et al., 2022). While contrail cirrus produced from global aviation have a net warming effect (Lee et al., 2021), contrails warm the atmosphere during night time and have a larger cooling contribution during day time (Frömming et al., 2021; Teoh et al., 2022). Hence, it is of great importance to understand their interaction with water vapor, atmospheric radiation, and natural cirrus. The atmospheric conditions necessary for the persistence of contrails and cirrus are the same and processes leading to cirrus formation often allow contrail formation, although contrails can also form ahead of cirrus. 90 Therefore, contrails and cirrus frequently coexist in the same regions (Schumann, 1996; Gierens, 2012). Contrail formation



and growth processes are driven by ambient water vapor causing a dehydration at the flight levels and enrichment at lower levels (Schumann et al., 2015). This influences the formation and life-cycle of natural cloudiness.

The way in which contrails and natural cirrus interact is an object of current research. Microphysical properties of contrails have been measured in many studies (e.g. Baumgardner et al. (1998); Voigt et al. (2010, 2011); Chauvigné et al. (2018); Li et al. (2022). Young contrails initially have ice crystal concentrations larger than $100 \ \mathrm{cm}^{-3}$ and ice crystal sizes of a few micrometers (Petzold et al., 1997; Gayet et al., 2012; Jeßberger et al., 2013; Kleine et al., 2018). Large tropospheric water vapor concentrations promote the growth of the ice crystals (Bräuer et al., 2021a, b), which leads to an overall higher optical thickness of the contrail (Wilhelm et al., 2022). If the supersaturation sustains, the contrails can live longer and both effects lead to a higher contrail radiative forcing at lower altitudes. During the transition from contrail to cirrus, the ice crystals grow by uptake of water vapor and the ice number concentration decreases due to mixing with ambient dry air (Schröder et al., 2000; Kübbeler et al., 2011; Voigt et al., 2017; Schumann et al., 2017; Grewe et al., 2017). Voigt et al. (2017) showed that the ice crystal number of aged contrail cirrus is still enhanced compared to natural cirrus.

Embedded contrails in natural cirrus and contrail cirrus have also been investigated with active and passive remote sensing. Tesche et al. (2016) found an increase in cloud optical thickness following the formation of a contrail in an existing cirrus. Following the same methodology, Marjani et al. (2022) focused on the number concentration and found an increase within a few hundred meters behind the aircraft and below its flight track. In line with airborne observations, Verma and Burkhardt (2022) simulated contrail formation within pre-existing cirrus and found that contrails increased the ice crystal number concentration of cirrus by a few orders of magnitude. Not only the ice number concentration is modified, but also the effective diameter. A reduction of ED was observed from measurements near flight corridors by Kristensson et al. (2000). Recent studies also showed a reduction in the effective radius of contrails embedded in natural cirrus (Wang et al., 2022; Li et al., 2022). However, depending on atmospheric conditions, when contrails are embedded in natural cirrus, the evolution of the contrail is perturbed by the cirrus and ice crystals from contrail and cirrus coexist, making it difficult to distinguish between them (Unterstrasser et al., 2017a, b).

Anthropogenic ice nucleating particles can influence ice crystal numbers and thereby the radiative properties of cirrus (Forster et al., 2021). INPs can increase the cirrus occurrence and can partially or completely suppress homogeneous freezing (Spichtinger and Cziczo, 2010; Kärcher et al., 2022). Regional differences between the concentrations, sources, and types of INPs may thus influence the cirrus microphysics. For instance, Beer et al. (2022) found lower concentrations of INPs in the less human-influenced high latitudes in the Northern Hemisphere in their model simulations. Aviation soot can also act as INP and indirectly affect the cirrus microphysical properties (Kanji et al., 2017; Urbanek et al., 2018; Groß et al., 2022). The emissions of aviation soot at cruise altitudes are the largest in mid-latitudes in the Northern Hemisphere and have the potential to significantly alter cirrus. However, their impact and even sign depend on the ice nucleation efficiency of aviation soot (Righi et al., 2021).

This work provides a statistical analysis of the differences in microphysical properties of cirrus measured during the CIRRUS-HL campaign (CIRRUS in High Latitudes) at high and mid-latitudes with the German research aircraft HALO (High Altitude and LOng range research aircraft). The measurements have been performed in June and July 2021 above Europe in



latitudes from 38 and 76° N. The data sampled during this campaign at mid-latitudes extend the ML-CIRRUS (Mid-Latitude CIRRUS) data set (Voigt et al., 2017) and incorporate high-latitude cirrus, which offers the rare opportunity to contrast mid- and high-latitude cirrus properties in the same season. The summer season data are of particular interest because in this period of the year the radiative effects of the cirrus are different at mid- (cooling) and high latitude (warming) (Hong and Liu, 2015).

In addition, the tropopause region above Europe was particularly supersaturated in summer 2021, which favours the formation of persistent contrails in the upper troposphere (Dischl et al., 2022).

An overview of the campaign is given in Sect. 2 and details on the instrumentation and data evaluation are explained in Sect. 3.1 and Sect. 3.2.1. In Sect. 3.2.2 we describe the approach of combining our cloud particle measurements from HALO with air parcel trajectories (Wernli and Davies, 1997; Wernli et al., 2016; Luebke et al., 2016) in order to relate cirrus properties to the

meteorological conditions at the location of cloud formation, as well as to classify the cirrus according to their origin (Krämer et al., 2016; Luebke et al., 2016; Wernli et al., 2016). An overview of the differences between mid- and high-latitude cirrus is described in Sect. 4.1, and in Sect. 4.2 we compare microphysical properties of cirrus sampled at mid- and high latitudes and in relation to the location of cloud formation in order to investigate a potential anthropogenic influence, mainly from aviation. The differences are also analysed depending on the cirrus origin type in Sect. 4.3. Finally, the results are further discussed in

Sect. 5 and a summary of the findings is given in Sect. 6.

## 2   The CIRRUS-HL campaign

Cirrus at mid- and high latitudes were measured with cloud probes on the research aircraft HALO during the CIRRUS-HL campaign in June and July 2021. The mission was based in Oberpfaffenhofen (Germany) and a total of 24 flights (22 science flights in cirrus) were performed between 24 June and 29 July 2021 covering regions in the Arctic, North Atlantic, and Central

Europe. The objective of the campaign was to investigate the cirrus formation and microphysical properties, as well as the radiative impact of cirrus at high latitudes and to contrast those with cirrus at mid-latitudes. To this end, we measured cirrus in different weather regimes, including warm conveyor belt cirrus, high pressure in situ origin cirrus, and convective cirrus. In addition, contrail cirrus and cirrus affected by aviation aerosol were targeted (Urbanek et al., 2018; Groß et al., 2022). In July 2021 the air traffic was reduced due to COVID-19 (Voigt et al., 2022) ($-46.5$ % available seats-kilometers according to

the International Civil Aviation Organization (ICAO) compared to 2019), but we still expect an influence of aviation in our observations of ML cirrus.

An overview of the flights during CIRRUS-HL is given in Fig. 1 and specific details of each flight are provided in Table 1. Targets specified as "ML" or "HL" cirrus stand for cirrus at mid-latitudes and cirrus at high latitudes, respectively. The flights targeted in convective cirrus (F12 and F15) are marked in green in Fig. 1 and have been excluded from the present analysis

since convection is not an objective of this study. A total of 34 h (17.7 h in mid-latitude cirrus, 7.8 in high-latitude cirrus and the remaining time in mixed-phase and liquid clouds) of in situ cloud particle measurements were achieved at different latitudes between 38 and 76° N at temperatures down to $-63$ °C and altitudes up to 14.3 km.



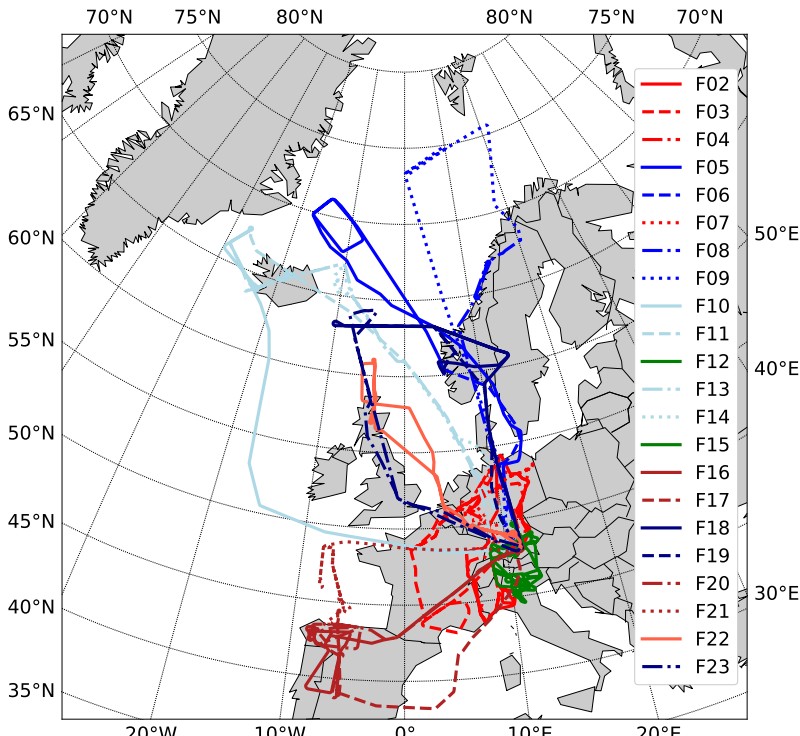

**Figure 1.** Map of the study region in the Arctic, North Atlantic, and Central Europe during the CIRRUS-HL mission in 2021. Each flight is labelled and drawn in a different colour and line style. Flights at high latitudes are indicated with blue colours, red colours represent flights at mid-latitudes and green is applied for flights in convection.

## 3 Methods

### 3.1 Instrumentation

During the campaign, HALO was equipped with a wide variety of in situ and remote sensing instruments to measure cloud particles, aerosols, trace gases ($H_2O$, $NO$, $NO_y$, $CO$, $CO_2$, $CH_4$ and $O_3$), radiation, and basic meteorological parameters (pressure, temperature, position, wind direction, and velocity). The data used in this study were collected by the Cloud Combination Probe (CCP) and the Precipitation Imaging Probe (PIP) mounted under the wings of the HALO aircraft and they are presented below. Detailed information about the other cloud probes and instrumentation installed for this campaign is given in Voigt et al.
(2017).



**Table 1.** Summary and objectives of the CIRRUS-HL science flights in June and July 2021. Measurement time, altitude, and temperature ranges are indicated only for the cirrus regime.

| Flight | Date | $t_{meas}$ [h] | Altitude [km] | Temperature [°C] | Targets |
|--------|------|--------|---------------|------------------|---------|
| F02 | 25 June 2021 | 0.7 | $[8.5, 9.8]$ | $[-49.5, -38]$ | ML cirrus, embedded contrails |
| F03 | 26 June 2021 | 1.67 | $[8.8, 11.4]$ | $[-56, -38]$ | ML cirrus, embedded contrails |
| F04 | 28 June 2021 | 2.09 | $[9.2, 12.2]$ | $[-60.9, -38]$ | ML cirrus, embedded contrails |
| F05 / F06 | 29 June 2021 | 1.9 / 1.35 | $[9.2, 12.5]$ | $[-62.7, -38]$ | HL and ML cirrus, embedded contrails |
| F07 | 01 July 2021 | 1.02 | $[8.7, 11.3]$ | $[-55.9, -38]$ | ML cirrus, embedded contrails |
| F08 / F09 | 05 July 2021 | 1 / 1.36 | $[9.2, 11.6]$ | $[-57, -39.6]$ | HL cirrus |
| F10 / F11 | 07 July 2021 | 0.5 / 0.78 | $[10, 13.8]$ | $[-54.3, -44.2]$ | HL cirrus, contrails |
| F12 | 08 July 2021 | 1.55 | $[9, 11.7]$ | $[-53.2, -38]$ | ML cirrus, convection |
| F13 / F14 | 12 July 2021 | 0.35 / 2.96 | $[8.8, 11.7]$ | $[-55.1, -38]$ | HL and ML cirrus, contrails |
| F15 | 13 July 2021 | 1.5 | $[9.5, 12]$ | $[-53.8, -38]$ | ML cirrus, convection, dust |
| F16 / F17 | 15 July 2021 | 0.36 / 1.17 | $[8.7, 14.3]$ | $[-61.6, -38]$ | in situ origin ML cirrus, contrail outbreak |
| F18 / F19 | 19 July 2021 | 1.21 / 0.3 | $[9.1, 11.8]$ | $[-60, -38]$ | HL cirrus, contrails, soot cirrus |
| F20 / F21 | 21 July 2021 | 0.66 / 0.35 | $[9.9, 13.6]$ | $[-56.4, -41.1]$ | in situ origin ML cirrus, contrails, CALIPSO overpass |
| F22 | 23 July 2021 | 1.06 | $[9.2, 11.5]$ | $[-57.2, -38]$ | HL cirrus, day-night, embedded contrails |
| F23 | 28 July 2021 | 0.69 | $[7.5, 11.9]$ | $[-52.9, -38]$ | HL cirrus and ML, CALIPSO overpass |

### 3.1.1 The Cloud Droplet Probe

The Cloud Droplet Probe (CDP) is part of the CCP from Droplet Measurement Technologies (DMT Inc., USA) (Baumgardner et al., 2011, 2017; Klingebiel et al., 2015; Weigel et al., 2016), and is a forward scattering probe that can size and count particles in a diameter range of $2-50\,\mu$m passing through the laser beam. The amplitude of the converted signal is related to the size of the particle applying Mie theory (Mie, 1908) (spherical particles) or T-matrix method (aspherical particles) (Waterman, 1971). While the size bin edges indicated by the manufacturer are calculated for spherical particles, ice crystals have different habits (Jang et al., 2022). The complexity of the ice crystals and the variety of aspect ratios make it difficult to obtain a standard solution with the T-matrix method. Borrmann et al. (2000) compared both methods for a FSSP-300 (Forward Scattering Spectrometer Probe model 300) and did not find substantial differences for small particles. Here, we only consider recordings of the CDP between 2 and 37.5 μm, we assume Mie theory and use the nominal bins from the manufacturer for simplification. The instrument was calibrated using glass beads (Lance et al., 2010).

The sample area of the CDP is $0.27\pm 0.025\,$mm$^2$ according to Klingebiel et al. (2015). The probe air speed (PAS) used to calculate the sampling volume of the CDP was measured by the pitot tube of the Cloud Aerosol Spectrometer with Depolarization Option (CAS-DPOL, Kleine et al. (2018)) mounted on the opposite wing at the same position as the CCP and not by the pitot tube of the CCP due to technical issues of the tube. With this quantity the particle number concentration was determined in a time resolution of 1 s.





The probe has two aerodynamic arms with pointed asymmetric tips upstream of the laser beam that reduce or prevent shattering in the CDP data (McFarquhar et al., 2007; Lance et al., 2010; Korolev et al., 2013). In addition, the interarrival time analyses from the CDP data recorded during CIRRUS-HL did not reveal the presence of shattering. McFarquhar et al. (2007, 2011) compared shattering effects in the CDP with other cloud probes (CAS-DPOL and FSSP) and found the CDP to be less susceptible to it. Owing to the low particle concentrations in cirrus ($< 1 \ \mathrm{cm}^{-3}$), errors induced by coincidence are considered negligible (Lance et al., 2010). The effects of the limited sampling volumes in the scattering probes were extensively explained by Krämer et al. (2020) in the appendix A2.3 with focus on the CAS-DPOL. In our case, the CDP has a lower N limit $\sim 0.025 \ \mathrm{cm}^{-3}$, when only one particle is recorded in one second. These so-called "single-particle-events" cause an increase in the frequency of the low N range of the CDP. We mitigated this effect by eliminating the single-particle-events due to the low statistics and inaccuracy of them.

### 3.1.2 The Cloud Imaging Probe and the Precipitation Imaging Probe

The Cloud Imaging Probe (CIPgs, gs for grayscale), the second part of the CCP, and the PIP are optical array probes (OAPs) (DMT Inc., USA) (Baumgardner et al., 2001; Weigel et al., 2016; Voigt et al., 2017)). Both probes consist of 64 photodiodes illuminated by a laser beam. When a particle passes through the laser beam, it projects a shadow on the diode array and each pixel is triggered if the light intensity falls below a threshold value (Knollenberg, 1970). The images are constructed by appending consecutive slices corresponding to the diode array state as the particle crosses the beam. The monoscale cloud probe PIP has only two light intensity levels and particles are recorded when at least one pixel of the image is $50\%$ obscured. On the contrary, the CIPgs is a grayscale probe and can distinguish between $0-25\%$, $25-50\%$, $50-75\%$, and $75-100\%$ nominal intensity levels. This functionality allows to record images already at $25\%$ dimming and adds details for the particle shape analysis. The threshold values used in this campaign were $35, 50$, and $65 \ \%$. We only considered images with at least one pixel at $50\%$ shadow intensity in order to assure a better agreement with the monoscale PIP and achieve a compromise between reducing the impact of out-of-focus particles (O'Shea et al., 2019, 2021) and keeping enough statistics. In addition, the size of the ice particles is calculated considering pixels with $50\%$ or higher shadow intensity. The size is calculated as the diameter of the minimum enclosing circle, which corresponds to the particle maximum dimension (Heymsfield et al., 2002). The CIPgs (PIP) has a nominal pixel size of $15 \ \mathrm{\mu m}$ ($100 \ \mathrm{\mu m}$) that allows to measure particles from $15-960 \ \mathrm{\mu m}$ ($100-6400 \ \mathrm{\mu m}$).

The PAS measured by the pitot tube attached to each instrument is applied as the sampling rate for image recording. The true air speed (TAS) is measured by the Basic Halo Measurement and Sensor System (BAHAMAS). In this case, the CIPgs was set to record the images at a higher fixed sampling rate than the real PAS (due to the technical issues with its pitot tube), which generates systematically artificial elongated particles. The images are corrected by applying as narrowing factor the ratio between PAS (measured by the CAS-DPOL) and the specified sampling rate. Further corrections of the images are also necessary due to alterations in the PAS data (e.g. icing on the Pitot tube), leading to distorted particles. Erroneous PAS measurement sequences are reconstructed using a PAS/TAS ratio and the deformed images can be corrected applying the ratio "corrected PAS" to "wrong PAS" (Weigel et al., 2016).



Particles were rejected if at least one of the end diodes was obscured and the sample volume was adapted accordingly, following the all-in method described by Knollenberg (1970) and Heymsfield and Parrish (1978). In addition, particles with only one pixel were excluded, as well as various pixel errors were identified and removed. Coincidence of particles was found to be irrelevant in the cirrus regime due to the low particle concentrations. The CIPgs has modified triangular tips that deflect the artifacts generated by the shattering of large ice crystals to avoid entering the sample volume. This modification is very efficient at reducing shattering effects (Korolev et al., 2013), however, an interarrival time analysis helped to identify and remove the remaining shattering events, which were not significant (Field et al., 2006). Although the PIP does not include the antishattering tips, we found less influence of shattering in the PIP than in the CIPgs, since the PIP can only detect particles from 50 µm and shattered particles are usually smaller. Baumgardner et al. (2017) estimated the error in sizing in $\pm 20\%$ for ice particles of diameters larger than $100 - 200$ µm and the error in concentration in $\pm 50\%$. For smaller particles, the error is larger and increases inversely proportional to the diameter.

## 3.2 Methodology

### 3.2.1 Cloud particle, relative humidity, and aircraft reference measurements

The size ranges of the three instruments overlap and thus it is necessary to calculate a combined size distribution. From 2 to 37.5 µm the CDP instrument is used, in the size range from 37.5 to 247.5 µm the CIPgs, in the range from 247.5 to 637.5 µm the mean concentration between the CIPgs and PIP is calculated, and from 637.5 to 6400 µm the PIP data were used to derive the total combined distribution. We based this strategy on a particle shape analysis of the CIPgs and PIP, which determined a size underestimation of complex ice crystals shapes in the lower size range of the PIP. An average particle distribution between 247.5 and 637.5 µm was found as the optimal solution to also compensate a slight size overestimation from the upper range of the CIPgs due to uncorrected out-of-focus ice crystals and the use of maximum dimension as sizing method, which leads to a higher overestimation when pixel errors are present.

The combined distribution of particle concentration in size bins for the whole cloud diameter range allows to calculate the microphysical cirrus parameters. To ensure completely glaciated clouds, we consider only measurements below $-38°C$ and calculate the N, ED, and IWC. We use the mass-dimension relationship from Heymsfield et al. (2010) to determine the IWC and the ED is computed as the ratio of the third to the second moment of the cloud spectrum (Parol et al., 1991; Schumann et al., 2011). For the calculations of IWC and ED we use the diameters that correspond to the center of the size bins.

In this study, we investigate the microphysical cirrus properties from 20 flights. As mentioned in Sect. 2, data from the flights in convection were removed. We apply a 2-s mean in order to improve the statistical significance of the low particle concentrations.

The pressure, temperature, and wind field measurements were performed by the BAHAMAS system operated by the DLR (German Aerospace Center) Flight Experiments department (Giez et al., 2021). This system also provides the basic aircraft position data of which we use longitude, latitude, and height. For the calculation of the $RH_i$, we use the water vapor mixing ratio measurements from the Sophisticated Hygrometer for Atmospheric ResearCh (SHARC), also developed and operated by



DLR Flight Experiments. The measurement range is between $2 - 50000\,\mathrm{ppm}$ and the overall relative and absolute uncertainty is $5\,\%$ and $\pm 1\,\mathrm{ppm}$, respectively (Kaufmann et al., 2015, 2018).

### 3.2.2 Air mass trajectories

Calculations of 10-day backward trajectories were performed from the flight tracks with the Lagrangian analysis tool LA-GRANTO (Wernli and Davies, 1997; Sprenger and Wernli, 2015) and using wind, temperature, and cloud fields from the operational European Centre for Medium-Range Weather Forecasts (ECMWF) analyses. Starting from the HALO flight paths, the hourly backward evolution of the IWC and liquid water content (LWC) along the trajectories was evaluated to estimate the time of cloud formation and distinguish between in situ and liquid origin cirrus following the approach described in Wernli et al. (2016). The formation point corresponds to the last time step along the trajectory before IWC $= 0$ occurs for the first time (going backward in time). However, if IWC $= 0$ only occurs at one hourly time step, we consider this as "noise", ignore this instance with IWC $= 0$ and repeat the criterion for the next point with IWC $= 0$. Once the formation point is determined, the corresponding in situ measurement is classified as of "liquid origin" if liquid water was present along the trajectory between cloud formation and the measurement (LWC $> 0$). If this was not the case and the air parcel only contained ice water during this time period, it is classified as "in situ origin" cirrus. Additionally to the cirrus origin classification, we also analysed the updraft speed along the backward trajectories to better understand the formation process.

## 4 Results

### 4.1 Overview and differences in microphysical properties of cirrus (N, ED and IWC) at mid- and high latitudes

The cirrus measurements are classified by latitude in order to investigate and compare microphysical properties of cirrus at mid- and high latitudes. For this first classification we consider the latitude at the measurement point. Data obtained at latitudes $< 60°\,\mathrm{N}$ are considered as mid-latitude (ML) cirrus and data collected at $\geq 60°\,\mathrm{N}$ are considered as high-latitude (HL) cirrus. This differentiation is somewhat arbitrary, as there is no universally accepted definition of the three general latitude zones. Perry (1987), for example, defined the mid-latitude in the Northern Hemisphere as the zone between $35°$ and $56°\,\mathrm{N}$ but the limits vary according to month and year. In Sect. 5.1 we discuss the variation of the latitude threshold and we find no major influence on the conclusions of this work (see also Fig. S3 in the Supplement).

Figure 2 shows an overview of the frequency distribution of the measured cirrus microphysical properties with respect to latitude. The normalized frequencies of occurrence allow the comparison between latitudes regardless of the number of observations in each bin. All together, we provide a cirrus data set from 41564 cloud samples in 2-s time resolution, of which 15211 samples are HL cirrus and 26353 samples are ML cirrus. Overall medians of N, ED and IWC (denoted with a tilde) for ML and HL cirrus are indicated in the Fig. 2 as well as in Table 2. We consider the median instead of the mean, since the mean is more affected by outliers. The overall median N for ML cirrus ($0.0086\,\mathrm{cm^{-3}}$) is higher than for HL cirrus ($0.001\,\mathrm{cm^{-3}}$) by



**Table 2.** Median, $25^{\text{th}}$ and $75^{\text{th}}$ percentiles of the microphysical properties N, ED and IWC in mid- and high latitudes during CIRRUS-HL.

| Latitude | N [cm$^{-3}$] | | | ED [$\mu$m] | | | IWC [g m$^{-3}$] | | |
| --- | --- | --- | --- | --- | --- | --- | --- | --- | --- |
| | 25% | Median | 75% | 25% | Median | 75% | 25% | Median | 75% |
| ML cirrus | 0.0007 | 0.0086 | 0.04 | 98 | 165 | 222 | 0.00037 | 0.0033 | 0.015 |
| HL cirrus | 0.00018 | 0.001 | 0.0092 | 146 | 210 | 329 | 0.00021 | 0.0019 | 0.011 |

an order of magnitude. The median ED in ML cirrus is 165 µm, smaller than that of HL cirrus, 210 µm and the median IWC do not differ much between ML and HL cirrus, as it represents a combined effect of N and ED.

Cirrus measured at mid-latitudes are characterized by higher $\widetilde{\text{N}}$ than those measured at high latitudes. $\widetilde{\text{N}}$ in HL cirrus takes mostly values between 0.0001 and 0.01 cm$^{-3}$. The central range of the ML cirrus measurements is shifted to values that are one order of magnitude larger ($0.001-0.1$ cm$^{-3}$). In addition, Fig. 2(a) also indicates that high particle number concentrations (N $> 1$ cm$^{-3}$) were more frequently observed at mid-latitudes. A linear correlation of the $\widetilde{\text{EDs}}$ is drawn in Fig. 2(b) and exhibits a slight and clear positive slope of $\approx 4$ µm per degree. The correlations of the $5^{\text{th}}$ and $95^{\text{th}}$ percentiles show that the

distribution of ED in ML cirrus is broader and smaller ED are more frequently observed with decreasing latitude. HL cirrus contain a lower IWC, however no significant correlation with latitude is noticed in Fig. 2(c). The range of values agree well with the findings by Schiller et al. (2008) for the Arctic and mid-latitudes. Voigt et al. (2017) also reported IWC between $10^{-6}$ and 0.2 g m$^{-3}$ with a high variability during the ML-CIRRUS campaign at mid-latitudes in spring.

     The cirrus properties from our measurements agree well with observations from previous campaigns. Brown and Francis

(1995); Heymsfield et al. (2010, 2013) found in general similar median values of N and IWC (mainly at mid-latitudes and the tropics). However, Brown and Francis (1995), for example, only included data from forward scattering probes. Here we use also the CIPgs and PIP, which have higher sample efficiencies. For this reason, we also measure lower IWC than 0.001 g m$^{-3}$. In general, our results also agree with those reported by Krämer et al. (2016); Luebke et al. (2016); Krämer et al. (2020) at mid-latitudes, but the larger EDs that result from our analysis are due to the addition of the PIP data and due to different sizing

methods applied in the calculation of the diameter from the CIPgs images. Here, we consider the maximum dimension of the ice crystals and not the area equivalent diameter because the maximum dimension diameter represents more accurately the spatial extent of ice crystals, which is key for radiative impact calculations. The lower N observed in HL cirrus compared to ML cirrus is consistent with the observations from Wolf et al. (2018). Marsing et al. (2023) showed averaged IWCs (0.003 and 0.005 g m$^{-3}$) of two case studies during the POLSTRACC (Polar Stratosphere in a Changing Climate) campaign in line with

our measurements at high latitudes.

     According to our measurements, the combination of higher N and smaller ED ice crystals is characteristic for ML cirrus. The cause of that can be manifold: more anthropogenic activities including aviation at mid-latitudes result in higher aerosol and ice nuclei loads in the upper troposphere. Gayet et al. (2004) also found higher N and lower ED in the more anthropogenically influenced Northern Hemisphere with respect to the more pristine Southern Hemisphere. Aviation-induced cirrus tend to have



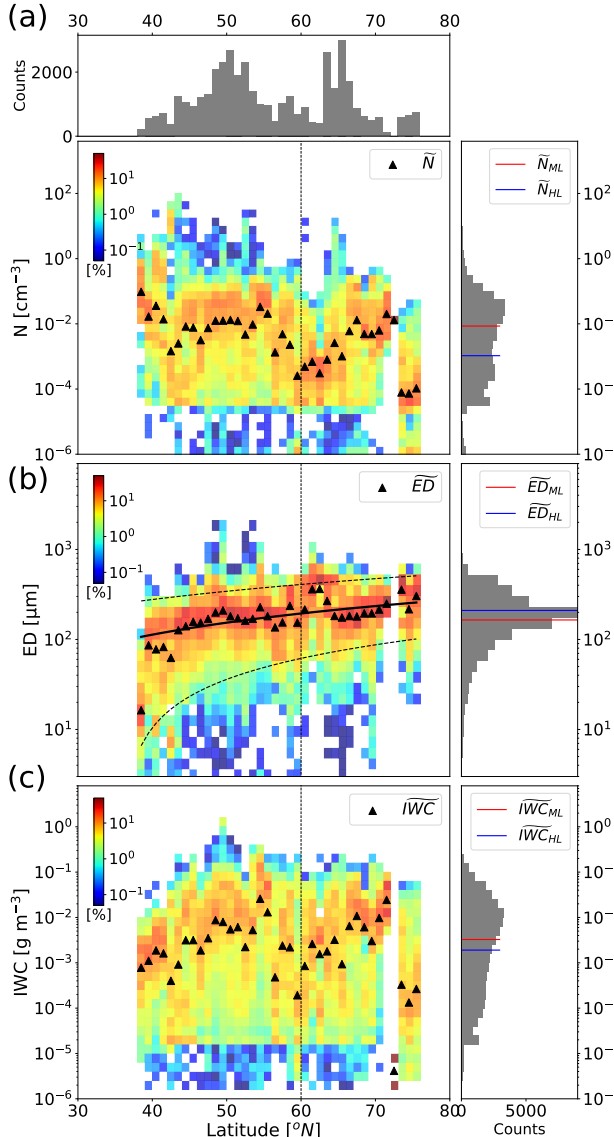

**Figure 2.** Normalized frequency distribution of (a) N, (b) ED and (c) IWC observations as a function of latitude of the measurement during CIRRUS-HL. Latitude bins are 1° wide and vertical bins are logarithmic. The colour code indicates the frequency of occurrence in percent per 1 degree latitude bins, normalized by the total counts per latitude bin. The vertical dashed line marks the threshold of 60° N for the differentiation of ML and HL cirrus. Triangular markers are medians per latitude bin ($\widetilde{N}$, $\widetilde{ED}$, $\widetilde{IWC}$). The top and right panels are histograms of the corresponding variables. The solid black line in (b) is a linear fit of the ED medians and the dashed lines are the corresponding linear fits of the 5[th] and 95[th] percentiles.



high number concentrations and lower EDs for several hours after their formation (Voigt et al., 2017; Schumann and Heyms-
field, 2017; Schumann et al., 2017). In general, it is difficult to distinguish aged contrails from thin natural cirrus unaffected by
aviation (Li et al., 2022) or embedded contrails in natural cirrus (Unterstrasser et al., 2017a, b), in particular, as the influence
of air traffic over Europe is omnipresent (Voigt et al., 2017; Schumann and Heymsfield, 2017).

In contrast to the ML cirrus, we often find that HL cirrus have lower N but larger ED, which could be attributed to the
uptake of the ambient water vapor by few INPs and the further growth of less ice crystals that allow larger sizes. The resulting
number concentration of the ice crystals depends on the temperature, the updraft and the number of INPs as well as their
capability to nucleate ice (Kärcher et al., 2006). Wolf et al. (2018) also found lower N at high latitudes and pointed at higher
concentrations of INPs at the mid-latitudes compared to the Arctic as a possible explanation. Beer et al. (2022) analysed global
model simulations under cirrus formation conditions and showed higher number concentrations of INPs (about $0.1$ cm$^{-3}$) and
higher number concentrations of newly-formed ice crystals (about $0.001$ cm$^{-3}$) between $30 - 60°$ N compared to $60 - 90°$ N.
However, a confirmation from airborne measurements of INPs is very challenging, as measurements in the cirrus regime with
temperatures lower than $-38$ °C are very scarce. DeMott et al. (2003, 2010) reported about different measurements of ice
particles residuals at temperatures $> -40$ °C and show a large variability in INP number concentrations depending on regional
and seasonal changes of the aerosol sources. Typically, mineral dust particles are found to have a high nucleation efficiency
(DeMott et al., 2003) and black carbon (BC) particles from anthropogenic sources also act as INP. In particular, the importance
of aviation soot particles as ice nuclei is not yet well determined. A number of studies investigated the aviation soot impact on
cirrus and showed a large range of possible model results, evidencing the uncertainty that exists in this regard (Hendricks et al.,
2011; Gettelman et al., 2012; Zhu and Penner, 2020; Righi et al., 2021).

A comparison between the temperature profiles of ED and N in ML (red) and HL (blue) cirrus is presented in Fig. 3 sorted
in $2.5$ °C temperature bins. We show temperature along the vertical axis as a representation of altitude. In Fig. 3(a) we observe
a clear decrease of ED with decreasing temperature for both ML and HL cirrus. Both profiles cover a similar temperature range
and correspond to altitudes between $8.5$ and $13.5$ km, with less measurements at high latitudes. The same decrease of ED
is also observed as a function of altitude. This effect has been already observed in previous studies and can be explained by
reduced atmospheric water content at lower temperatures (or higher altitude), together with the different altitudes at which in
situ and liquid origin cirrus exist (Luebke et al., 2016; Wernli et al., 2016). With increasing altitude (decreasing temperature),
cirrus origins change from cirrus dominated by the liquid origin regime with larger ice particles to in situ origin cirrus with
smaller particles (see also Voigt et al. 2017). This is expected due to the low-altitude origin of the liquid origin cirrus (Luebke
et al., 2016). Larger particles can also be found at higher temperatures (lower altitude) resulting from sedimentation processes.

Between $-50$ and $-38$ °C we find a remarkable difference between the 25[th] and 75[th] percentiles in the ED between ML and
HL cirrus. This fact probably indicates a more dominant influence of liquid origin cirrus at the high latitudes with enhanced
sedimentation as a result of the larger particles at these latitudes. From the temperature profiles we can confirm the larger EDs
observed at high latitudes in the general picture of Fig. 2(b). Higher ED in HL cirrus compared to ML cirrus are observed in
almost all temperatures. To confirm this finding, we use the U-test according to Wilcoxon, Mann and Whitney (mannwhitneyu
function from the Python sub-package scipy.stats) to assess whether there is a statistically significant difference between the



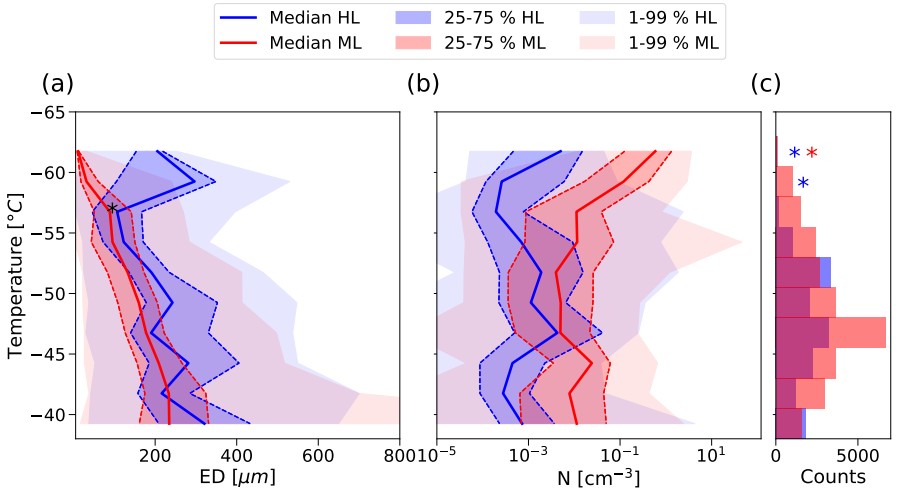

**Figure 3.** Distribution of (a) ED and (b) N observations sorted in temperature bins of 2.5 °C for ML (red) and HL cirrus (blue) and (c) histogram of the total frequency of observations per temperature bin for each group. The thicker solid lines correspond to the median values per temperature bin. The dark shaded areas between the dashed lines comprise the 25$^{\text{th}}$ and 75$^{\text{th}}$ percentiles of the population. The lighter shaded areas mark the extension between the 1$^{\text{st}}$ and 99$^{\text{th}}$ percentiles. Black asterisk indicates low statistically significant difference between medians. Blue and red asterisks indicate low statistics in HL and ML data, respectively (see text for further details).

ML and HL medians per temperature bin for ED (and also for N). The null hypothesis that the distributions are equal was rejected for all cases with p-value $< 10^{-6}$, except for the ED medians between $-58$ and $-55.5$ °C with a p-value $= 0.028$. Therefore, we conclude that the observed differences are statistically significant. Between $-45$ and $-38$ °C at mid-latitudes, we observe high ED values in the 1$^{\text{st}}$ to 99$^{\text{th}}$ percentiles. These events are connected to isolated convective systems over Germany with an enhanced growth of the ice particles.

Contrary to the ED, N does not show a clear tendency with decreasing temperature in either case. However, the N in ML and HL cirrus differ at all temperatures by about an order of magnitude, more pronounced at lower temperatures. However, the ML cirrus coldest temperature bin, which mostly contains high N, has reduced statistics with a total of 119 2-s samples measured from four different cloud sequences. Between $-62$ and $-55$ °C there is an enhancement of N in ML cirrus associated with a steeper decrease in ED, which is probably connected to contrail formation. According to Bräuer et al. (2021a), contrail

conditions are favorable between $-50$ and $-60$ °C. Here contrails have the highest extinction coefficients. This corresponds to altitudes between 9.5 and 11 km, where short- and medium-range commercial flights are typically located. Contrails are frequently present in ML cirrus above Central Europe. Here, we mostly find high N and low ED around $-55$ and $-60$ °C in our data set. High N values ($> 1 \ \text{cm}^{-3}$) in the other temperature ranges are rare and represented by the outliers, which do not impact the medians. Measurements of HL cirrus below $-55.5$ °C are less representative with 210 counts in total. In particular,

between $-63$ and $-60.5$ °C we find 22 2-s consecutive samples from the same cloud sequence. Between $-60.5$ and $-58$ °C





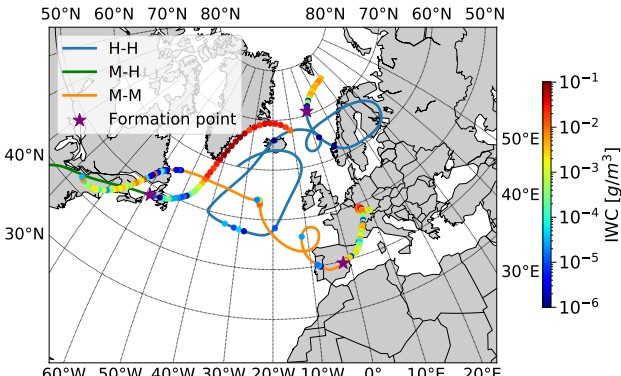

**Figure 4.** IWC (colour coded) along three example trajectories from the cirrus classification: formation and measurement at high latitudes (H-H, blue), formation at mid-latitudes and measurement at high latitudes (M-H, green), and formation and measurement at mid-latitudes (M-M, orange). The formation point is indicated with a purple star.

three different cloud sequences were probed with a total of 40 counts. Fluctuations in ED in HL in these temperature ranges in Fig. 3(a) can thus be explained by the reduced statistics.

## 4.2   Relating ice cloud microphysical properties to the location of cloud formation

The main approach of this study is the analysis of cirrus microphysical properties depending on their latitude. So far, we have classified the data in ML and HL cirrus using a latitude threshold at $60°$ N regarding the measurement point. Now, we investigate the meteorological conditions at the time of cirrus formation and how they affect the observed microphysical properties. Following this strategy, we obtain four categories: cirrus measured and formed at mid-latitudes (M-M), cirrus formed at mid-latitudes and measured at high latitudes (M-H), cirrus formed at high latitudes and measured at mid-latitudes (H-M), and cirrus formed and measured at high latitudes (H-H). Examples of backward trajectories for each group are shown

in Fig. 4. The H-M cirrus is excluded from the analysis, as it only contains 36 data points, all of them just at the limits of the boundaries (see Fig. 5). In Fig. 4, the points with IWC $> 0$ in the trajectory before the purple star indicate a previous formation of ice within the air parcel, which evaporated later.

Figure 5 shows the distribution of 2-s measurements for the four categories delimited by the $60°$ N threshold and colour coded according to ED. The three important categories are M-M in the bottom left, M-H in the bottom right, and H-H in the

upper right. Using this categorization, we separate the measurements previously classified as HL cirrus into cirrus formed at high latitudes (H-H) and cirrus formed at mid-latitudes and transported to high latitudes (M-H). The number of data points found in each category (4450 for H-H and 10753 for M-H) indicates that the category M-H contributes more frequently to the Arctic cirrus observed in our measurement campaign.

It becomes evident that the distribution of the measured ED is connected with the latitude at the measurement point. Reddish

colours represent high ED of up to $1000\,\mathrm{\mu m}$ and the dark blue dots represent EDs of about $10\,\mathrm{\mu m}$. Higher occurrences of smaller




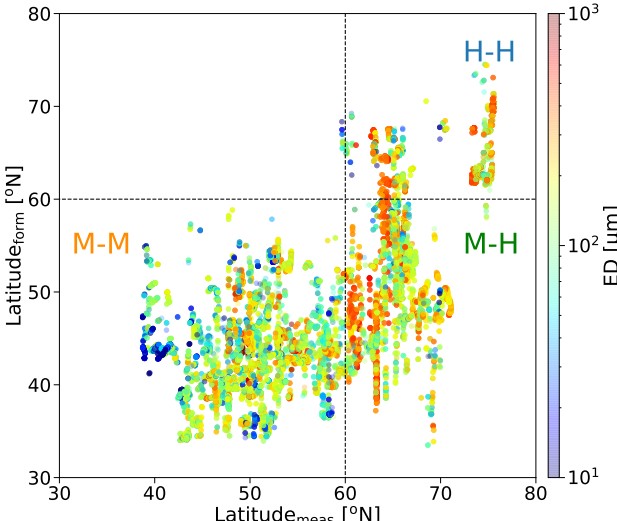

**Figure 5.** Correlation of latitude at the formation and latitude at the measurement for 2-s measurement points. The colour indicates the associated ED. Four regions are delimited with black dashed lines and define the groups: H-H, M-H and M-M (see text for details).

particles are measured in ML cirrus and higher occurrences of larger particle sizes are found in HL cirrus. The differences are smaller when looking at the latitude of formation of cirrus measured in high latitudes. We can conclude that the measurement location has a larger influence on the particle size than the formation region.

In order to investigate in more detail how the cirrus properties depend on their source region, Fig. 6 illustrates normalized
frequency distributions of N, ED and $RH_i$ of the three categories. The maximum of the N distribution of the M-M cirrus lies between 0.01 and 0.1 $cm^{-3}$. In contrast, the highest probability in the H-H cirrus is shifted to significantly lower values between 0.0001 and 0.001 $cm^{-3}$. The M-H cirrus distribution exhibits an intermediate behaviour. The medians reflect the same observation, with 0.0086, 0.0018 and 0.0004 $cm^{-3}$ for the M-M, M-H and H-H cirrus, respectively. The ED profiles in Fig. 6(b) show the opposite trend, as do the medians (164, 206 and 225 μm for M-M, M-H and H-H cirrus, respectively).

In addition, Fig. 6(c) shows the measured $RH_i$ for the three categories. In general, the tropopause region in summer 2021 had a high occurrence of supersaturation both at mid- and high latitudes. We observe a clear difference in the $RH_i$ distribution with lower $RH_i$ with a median of 107 % in M-M cirrus and higher values in the H-H cirrus with a median of 125 %. This can be explained by the lower ice crystal concentrations at high latitudes due to a reduced availability of INPs. The relative humidity is not sufficient to homogeneously nucleate new particles and, instead, the small number of ice crystals available
takes up the abundant water vapor increasing their size. On the contrary, water vapor at mid-latitudes is rapidly consumed by the availability of many ice particles present in the upper troposphere, which only leads to a moderate growth.

The M-M cirrus is 57 % of liquid origin and 43 % of in situ origin in our data set. Regarding HL cirrus, on the one hand we find that 90% of the measurements corresponding to the M-H cirrus have a liquid origin with a larger cloud age. On the other hand, 86% of the H-H cirrus data points are classified as in situ origin cirrus. Therefore, M-H cirrus were formed from



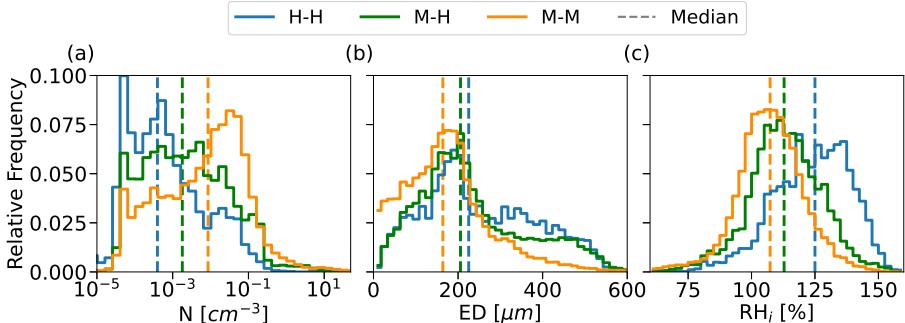

**Figure 6.** Relative frequency distributions of (a) N, (b) ED and (c) $RH_i$ of the cirrus formed and measured at high latitudes (H-H, blue), cirrus formed at mid-latitudes and measured at high latitudes (M-H, green) and cirrus formed and measured at mid-latitudes (M-M, orange). The distribution of N is given in logarithmic bins. Linear bins of $15\,\mu m$ and $2.5\%$ width have been chosen for ED and $RH_i$, respectively. The corresponding median values of each variable and category are depicted with dashed lines.

the liquid phase in a ML environment rich in nucleating particles. Here many small particles were nucleated in a first stage. Then, the air parcels are advected northwards to high latitudes, bringing the clouds into a highly supersaturated atmosphere allowing the formed ice particles to grow. Cirrus measured at high latitudes can be clearly distinguished from ML cirrus by the bimodality of the ED distribution. The bimodality of the ED distribution of M-H and H-H cirrus can be understood by two different processes: higher ice crystal growth at high latitudes due to the increased supersaturation leading to the sedimentation

of large ice crystals from the upper layers or different nucleation times during their lifetime with an enhanced growth of the larger mode due to fewer INPs and high supersaturation.

These findings together clearly show an influence of the ML origin in the development of cirrus at high latitudes, conserving the microphysical properties of the ML cirrus while being transported to higher latitudes. In this way, the formation region influences and defines the initial ice crystal properties, which then mix or are modified during their lifetime due to the different

atmospheric conditions. Thus, the largest fraction of clouds observed during CIRRUS-HL originate from the mid-latitudes and through long range transport modify the HL cirrus. Opposite influences from high to mid-latitudes have not been observed.

We can conclude that latitudinal differences in microphysical parameters are better understood when looking at both the location of the ice crystals formation and the location of the measurement. The region where the ice crystals are formed influences the initial cirrus properties, in particular the initial N. The latitude at which the cloud particles were measured

determines the resulting state of the measured ice crystal properties, strongly influenced by the $RH_i$ throughout the cirrus life cycle and mainly affects the measured ED. Both processes are largely influenced by the updraft on small and large scales, which are discussed below.



### 4.3  Liquid and in situ origin cirrus at mid- and high latitudes

Some of the differences in the microphysics of the cirrus measured during CIRRUS-HL discussed in Sect. 4.1 could be at-
tributed to a greater or lesser presence of in situ or liquid origin cirrus at mid or high latitudes. Here, we classify the measure-
ments making use of the information on LWC and IWC along the backward trajectories after the formation of ice in the cloud,
as described in Sect. 3.2.2. We obtain four groups with this division: in situ origin HL cirrus, liquid origin HL cirrus, in situ
origin ML cirrus and liquid origin ML cirrus.

The frequency of measurements of each group as well as their microphysical properties in terms of ED and N are presented
in Fig. 7. Liquid origin cirrus were frequently measured at mid-latitudes and were also the dominant type at high latitudes.
Liquid origin HL cirrus were observed almost as frequently as the in situ origin ML cirrus. In situ origin HL cirrus, in turn, were
the least measured. However, isolated rare events are not well represented in the diagrams due to the calculation of contours by
interpolation.

Larger particles than the total ED median value ($180\,\mu m$) were found in both cirrus types at high latitudes, with larger values
for liquid origin cirrus. In situ origin HL cirrus were rather thin, with N often under $0.004\,cm^{-3}$, the median value of all data.
The ED distribution of both cirrus types at high latitudes exhibits two distinct maxima, as seen before in the M-H and H-H
cirrus in Fig. 6(b). Large EDs representative of liquid origin cirrus are present in mid-latitudes with some outliers of high ED
and high N in convective cells involving also precipitation particles. However, smaller EDs than the median, even for the liquid
origin cirrus, are more frequent at mid-latitudes.

The observations at high latitudes generally agree well with the measurements in the Arctic from Wolf et al. (2018). However,
high number concentrations are not found in our in situ origin HL cirrus in contrast to Wolf et al. (2018). The reason is that
their measurements of Arctic in situ origin cirrus were dominated by homogeneous nucleation events driven by high updraft
motions. Our analysis of the updraft speed along the backward trajectories and the measured vertical velocities indicate rather
low vertical velocities. Specific information on the updraft speeds along the backward trajectories of each of the four categories
can be found in Fig. S2 of the Supplement. In addition, orographic cirrus were not a target of the campaign and therefore we do
not expect mountain wave cirrus with high number concentrations. We assume that mainly heterogeneous nucleation defined
the cloud formation in our data set. However, it should be noted that the temporal resolution of the parameters calculated along
the trajectories is one hour and therefore, effects on the updrafts from small-scale fluctuations and turbulence are not taken into
account here. The prevalence of heterogeneous freezing was also shown by Froyd et al. (2022) at mid- and high latitudes in the
Northern Hemisphere during spring and summer due to the suppression of homogeneous nucleation by intense dust emissions.
Saharian dust emissions were forecasted and reported during the campaign, but its influence on cirrus formation has not been
investigated in this work.

Greater differences are found among the in situ origin cirrus. While in situ origin ML cirrus have higher N with smaller
EDs mainly between 70 and $250\,\mu m$, in situ origin HL cirrus are mostly composed by low number concentrations of large
particles with ED between 150 and $450\,\mu m$. As for the ED medians, the values of liquid origin cirrus change from $188\,\mu m$
at mid-latitudes to 220 at high latitudes, while the in situ origin cirrus does it from 128 at mid-latitudes to $189\,\mu m$ at high

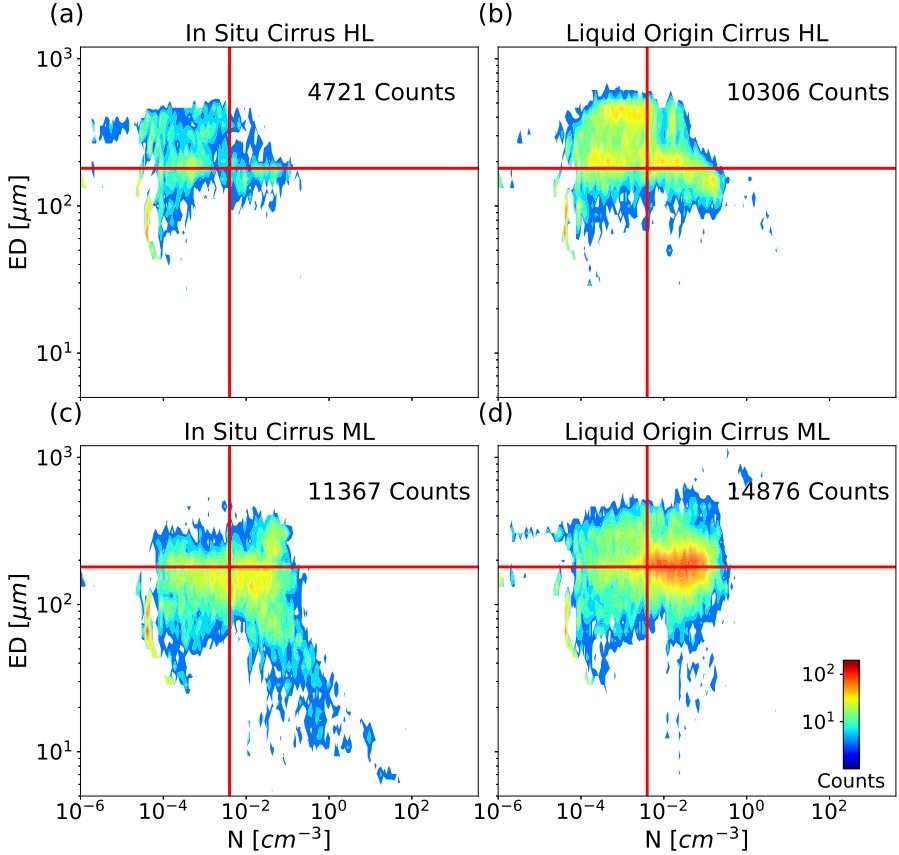

**Figure 7.** Frequency of ED observations in counts as a function of N. The observations are separated in (a) in situ origin HL cirrus, (b) liquid origin HL cirrus, (c) in situ origin ML cirrus, and (d) liquid origin ML cirrus. The method to draw the contours uses a marching squares algorithm. Vertical and horizontal solid red lines indicate overall median values of ED and N, respectively, and are shown to better illustrate differences. The total number of 2-s data points are given on the upper right corner of each panel.

latitudes. We also draw the readers' attention to a feature in the bottom right square of Fig. 7(c) of in situ origin ML cirrus, which differs from the in situ origin HL cirrus observations. Although we did not quantify the contribution, we identified high concentrations of small particles connected to young contrails, aged contrails, contrail cirrus or embedded contrails in natural
cirrus. This feature is also weakly present in Fig. 7(d) of liquid origin ML cirrus but substantially less pronounced. We discuss in more detail the aviation influence in Sect. 5.2.

Our previous analysis in Sect. 4.1 already suggested that ML cirrus are more affected by contrail cirrus. As Li et al. (2022), we observe that in situ origin cirrus are more strongly influenced by contrails than liquid origin clouds. Liquid origin clouds are formed at lower altitudes where liquid droplets and ice crystals can coexist. Ascending motion brings the cloud to lower
temperatures with a subsequent increase in relative humidity. The Wegener-Bergeron-Findeisen (WBF) process leads to the





growth of the ice crystals by water vapor uptake from the evaporated liquid droplets and finally to the complete glaciation of the cloud (Korolev, 2007; Costa et al., 2017). The evolution of the liquid origin cirrus has sufficient time to consume the available water vapor when the cloud reaches the cirrus regime. Li et al. (2022) found that the Schmidt-Appleman criterion for contrail formation (SAC) is mostly not fulfilled in liquid origin clouds due to the warmer temperatures at which they are
located. Therefore, it is more difficult to find the appropriate conditions for a contrail to develop within a liquid origin cloud.

On the contrary, in situ origin cirrus form directly at high altitudes, where air traffic is also present. If we imagine the scenario in which an in situ origin cloud starts forming heterogeneously by the deposition of water vapor on INPs and an aircraft exhaust jet is introduced, the natural ice particles to be formed would be substituted by the freshly formed contrail affecting severely the resulting microphysical properties. Contrail formation within cirrus can increase the cirrus ice crystal number by a few
orders of magnitude (Schröder et al., 2000; Voigt et al., 2017; Schumann et al., 2017), in particular if the pre-existing cirrus has high ice water content and the supersaturation is low (Verma and Burkhardt, 2022). In this sense, we can expect an influence of contrail formation and growth on the formation process of in situ origin cirrus at cruise altitudes by an increase in the N and reduction in ED.

Additional information on the properties of the cirrus groups is given by their particle size distributions (PSDs). Knowing
the ice crystal size distribution is vital in order to derive the radiative impact of cirrus. The representation of PSDs of the four groups is not a trivial task in this case. We usually find in other publications the mean PSD as representative of a considered data set. However, our campaign deals with a wide variety of situations, including outliers, which were in fact infrequently observed but strongly dominate the mean. These events are, in our case, contrails and convection encounters. This implies that the mean PSD fails to identify the averaged behaviour in the group and the use of medians appears to be a better choice for this
case. However, due to the low concentrations in cirrus and the lower sampling efficiency of the CDP compared to the CIPgs and PIP, the lower bins are frequently empty in the 2-s sampling rate and do not allow a median value calculation. Therefore, we increase the averaging intervals to 180 s to calculate the median concentration in each size bin. Sensitivity analyses were performed to select the 180 s averaging interval.

The calculated PSD median values are shown for each cirrus group in Fig. 8(a) and the means in Fig. 8(b) are also included
for comparison. The median concentrations in each bin of the liquid origin cirrus are higher than for the in situ origin cirrus for particles > 20 μm at both mid- and high latitudes and the former covers a larger size range than the latter, consistent with previous observations (Krämer et al., 2016; Luebke et al., 2016). The median PSDs of liquid and in situ origin ML cirrus have higher concentrations than HL cirrus. Both ML cirrus profiles have the same behaviour in the smaller bin range (2 − 100 μm) and they differ in the larger sizes. The liquid origin cirrus reveal a second mode, slightly less pronounced than at high latitudes,
and the in situ origin ML profile decays fast without a second mode.

As for the mean values, we detect two clear outlying features. Firstly, the size range in liquid origin ML cirrus is larger, which is a consequence of the isolated convective events that were encountered. Secondly, the large number concentrations of small crystals for the in situ origin ML cirrus clearly shows that the microphysics of in situ origin ML cirrus were affected by contrails. The difference in the concentration of the lower bins (from 2 to 10 μm) between the median and the mean is
about two orders of magnitude. Then, the mean concentration decays and almost converges with the median at about 40 μm.




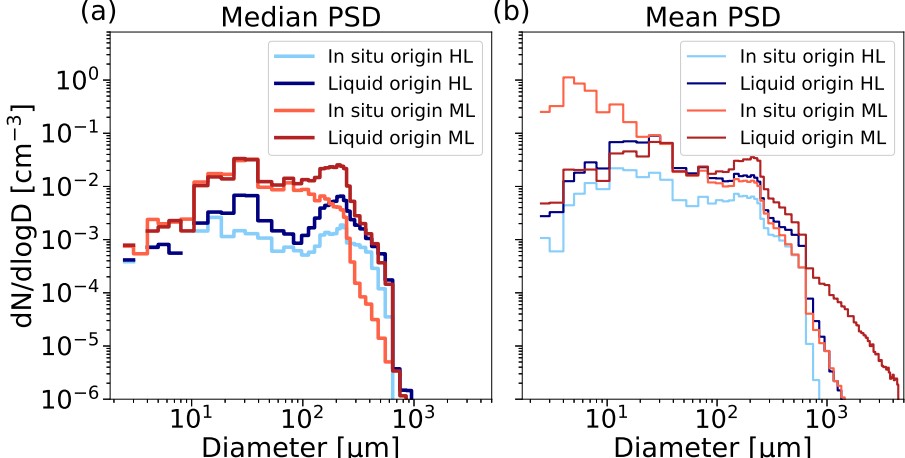

**Figure 8.** (a) Median and (b) mean particle size distributions (PSD) of the groups: in situ origin HL cirrus (light blue), liquid origin HL cirrus (navy), in situ origin ML cirrus (light red) and liquid origin ML cirrus (dark red). Median PSDs are indicated with thick lines and means with thinner lines.

In general, the median PSDs exhibit lower concentrations than the means, in all bins for the four groups, which is expected due to the right skewness of the lognormal distribution. The mean and median show larger differences the more outliers there are in the distribution. The PSDs of the in situ and liquid origin HL cirrus show a weaker influence of outliers, as the medians profiles do not differ much from the means.

The increased humidity in the upper troposphere during the campaign period can contribute to more persistent contrails which can be particularly warming (Wilhelm et al., 2022; Teoh et al., 2022; Wang et al., 2022). However, the potential contrail cirrus cover is the lowest during summer over Europe (Dischl et al., 2022). The increase in air temperature in summer hinders the formation and persistence of contrails and their radiative effect is mostly cooling because of the increased sun hours in summer in the North Atlantic corridor (Teoh et al., 2022). Our study confirms a low frequency of direct observations of contrail

encounters. Even though they do not dominate the campaign observations and therefore are not representative of the general picture, the mean PSD of in situ origin ML cirrus shows how strongly contrail encounters can affect the cirrus microphysics with an enhancement of smaller particles. More importantly, our analyses indicate that higher number concentration and smaller particles indeed describe the microphysical properties of ML cirrus compared to lower concentrations and larger ice crystals at the more pristine high latitudes.



# 5 Discussion of limitations

## 5.1 Influence of the method

The selection of the latitude threshold of 60 °N is a critical aspect in our study, as it determines the differentiation between mid- and high latitudes and, at the same time, our cirrus classification. We performed a sensitivity study and varied the threshold between $55 - 65$ °N (see Fig. S3 in the Supplement). Here, we show that the variation of the threshold does not alter the existence of the described differences between the three cirrus groups, even though it influences the absolute values of the medians. In addition, Fig. 2(b) shows a continuous tendency of increasing effective diameter from mid- to high latitudes, which shows that the differences between ML and HL cirrus are continuous and do not depend on the threshold.

The classification of cirrus origins of our in situ measurements is susceptible to uncertainties as trajectories were initiated along the flight track every 10 s, which corresponds to $\approx 2$ km in horizontal scale. We assume that cirrus measured in this interval have the same origin. In addition, the model parameters calculated along the trajectories are averages of grid points in a $\approx 50 \times 50$ km² domain that does not capture the smallest local variations. We use the vertical motion along trajectories to separate processes connected to our measurements. However, our trajectory approach excludes the analysis of the impact of small-scale temperature and wind fluctuations on cirrus formation. The accuracy of the approach is difficult to assess as the associated uncertainty is very in-homogeneous, it depends on the location and the meteorological situation. Nevertheless, we yield convincing results from our classification method, which are in line with previous publications (Krämer et al., 2016; Luebke et al., 2016; Wernli et al., 2016; Krämer et al., 2020).

## 5.2 Aviation influence

Even though the conditions for contrail formation are not as favourable in summer compared to winter (Dischl et al., 2022) and even though the air traffic was still reduced in 2021 due to COVID-19 (ICAO, 2022; Schumann et al. 2021; Voigt et al. 2022), we could identify an influence of contrail formation and interaction with natural cirrus in our data set. We did not focus on a methodical approach to differentiate contrails or contrail cirrus from natural cirrus (Li et al., 2022) but we considered as non natural cirrus measurements with N > 0.1 cm⁻³ and ED > 40 μm (lower right section of Fig. 7(c)). In general we did not observe high updraft speeds during the campaign, neither from the trajectories nor from the in situ measurements (see Fig. S2 in the Supplement). In particular, the events with high N and low ED were also not associated with high updraft speeds that could lead naturally to strong cooling rates and an outburst of small ice crystals. Additionally, we supported the analysis with flight reports and contrail cirrus predictions. With this information we are able to indicate the existence of an aviation influence and identify this influence on the properties of the described cirrus groups.

## 5.3 Representativeness

This study analyses the differences in the properties of the cirrus that we measured during the CIRRUS-HL campaign according to the latitude. The results are representative of summer 2021, of the regions measured over the Arctic, North Atlantic and



Central Europe and of the meteorological situations observed during the campaign. Under this context, our cirrus observations cover a wide range of latitudes between 38 and 76° N and temperatures between −38 and −63 °C. The statistical tests we performed indicate a sufficient statistical significance of the differences that we found in these ranges but conclusions drawn on the coldest temperatures (between −58 and −63 °C for HL cirrus and −60.5 and −63 °C for ML cirrus) have to be taken

cautiously as the number of observations is limited and the samples are mainly from the same cloud sequence. This implies that the observations are correlated to some extent, as they are measured at approximately the same temperature and height. However each 2-s sample is horizontally separated by ≈ 800 m and thus the cloud properties are not necessarily connected. In any case, our measurements can be combined with future observations to build a larger data set and gain a broader perspective.

## 6   Summary

In this study we provide new insights into the microphysical properties of the rarely observed Arctic cirrus and support our analysis by backward trajectories of the cirrus air parcels to investigate cirrus formation and evolution. Even though other campaigns have been previously performed in this region (Schiller et al., 2008; Heymsfield et al., 2013; Krämer et al., 2016; Wolf et al., 2018; Marsing et al., 2023), in situ data on the full range of cirrus particle sizes remain limited at high latitudes. Contrary to other campaigns at high latitudes, we also measured cirrus at mid-latitudes during the same mission with the same

instrumentation, which provided us with a unique opportunity for comparison. The main findings are stated and discussed in the following:

- High-latitude cirrus measured during summer 2021 are characterized by lower concentrations and higher effective diameters than mid-latitude cirrus. Similar differences in high-latitude cirrus were also observed by Wolf et al. (2018) compared to measurements at mid-latitudes (Krämer et al., 2016; Luebke et al., 2016). As Wolf et al. (2018), we also

suggest the reduced amount of available ice nucleating particles at high latitudes as a possible explanation for these findings. In situ measurements of INPs from Arctic cirrus are urgently needed in order to shed more light on this issue.

- The upper troposphere in the summer 2021 was characterized by high ice supersaturation, especially in high-latitude cirrus ($\widetilde{RH_i} \sim 125\%$). Contrary to wintertime observations in the Arctic by Wolf et al. (2018), which included fast updraft lee wave cirrus, we often analysed in situ origin high-latitude cirrus formed at lower updrafts. Different from

Wolf et al. (2018), the analysis of updraft speeds along the backward trajectories indicates that heterogeneous freezing dominates cirrus formation during this campaign.

- The Arctic cirrus originate from two processes. First, the cirrus formed and measured at high latitudes consisted mainly of heterogeneously formed in situ origin cirrus nucleated in slow updrafts. Few ice nucleating particles and high relative humidity over ice produced low ice number concentrations and larger effective diameters. The high-latitude cirrus that

had formed at mid-latitudes were dominated by liquid origin cirrus formed at mid-latitudes with higher number concentrations and transported to high latitudes with a subsequent growth of the ice crystals. This category is a mixture of the cirrus properties at mid- and high latitudes and represents how the mid-latitudes influence the properties of cirrus at high



latitudes. Both cirrus categories contribute to the larger effective diameter and smaller number concentration measured at high-latitude cirrus compared to mid-latitude cirrus.

- We discuss the selection of the latitude threshold of 60 °N and the possible influence on the results. We found slight variations on the microphysical properties of the investigated cirrus groups but the differences between them remain consistent, regardless of the chosen threshold.

    - Although both liquid and in situ origin cirrus at mid-latitudes exhibit lower effective diameters more frequently than at high latitudes, the differences between in situ origin cirrus at mid- and high latitudes are more substantial. We point out

the contribution of contrail formation within the in situ origin cirrus in flight corridors at mid-latitudes to the differences observed between the in situ origin cirrus in mid- and high latitudes (Voigt et al., 2017).

    - We discuss the influence of aviation in the data set and its representativeness. Our results show a possible contribution of contrail formation in the changes of the cirrus properties.

    - We introduce the median particle size distribution as a more suitable alternative representation than the mean for our

case, in order to obtain particle size distribution that represent the typical values of sample groups with high variability and strong outliers.

The introduction of the new cirrus classification by taking into account not only the measurement location but also the location at cirrus formation is an important insight for future studies. As we show from our measurements, part of the cirrus considered as high-latitude cirrus are actually influenced by mid-latitude air masses that change their properties. In situ mea-

surements of ice nucleating particles in the cirrus regime are much needed to clarify the cause of the different cirrus properties at high latitudes compared to mid-latitudes. Combining in situ observations with modelling studies could provide further insights into the underlying processes influencing cirrus formation and cirrus cloud properties. The present study aims to contribute and enhance our knowledge of cirrus formation processes and their microphysical properties in high and mid-latitudes in order to compile a database for studies on their climate impact.

*Data availability.* Processed data from the CIRRUS-HL campaign are available at the HALO data base at https://halo-db.pa.op.dlr.de/.

*Author contributions.* EDC conducted the analysis and wrote the manuscript. TJW and CV supervised the study, provided intensive feedback on the manuscript and coordinated the CIRRUS-HL mission. EDC, TJW, VH, JL, MZ and CV conducted the in-flight measurements. TJW, AA, VH, SK, MK, JL, NS and CV supported the particle data evaluation. HW performed the backward trajectories calculation. VG supported the statistical analysis. All authors contributed to and commented on the manuscript.



*Competing interests.* At least one of the (co-)authors is a member of the editorial board of Atmospheric Chemistry and Physics. The peer-review process was guided by an independent editor, and the authors have also no other competing interests to declare.

*Financial support.* This work is supported by the German Research Foundation within SPP-1294 HALO (grant nos. VO1504/6-1, VO1504/7-1 and VO1504/9-1) and TRR 301 (Project-ID 428312742).

*Acknowledgements.* We would like to express our gratitude to the flight department and weather forecast teams for their excellent support
during the campaign.



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
