# Peer review of "Differences in microphysical properties of cirrus at high and mid-latitudes"

_EGUsphere, 2023_

## Referee Comment (RC1)

Review of "Differences in microphysical properties of cirrus at high and mid-latitudes", EGUsphere 2023-374, by Castro, Jurkat-Witschas, Afchine, Grewe, Hahn, Kirschler, Kramer, Lucke, Spelten, Wernli, Zoger, and Voigt.

This is an interesting and novel study that uses data from the CIRRUS-HL field program that collected airborne observations at latitudes from about $35^0$ to $60^0$N and $60^0$ to about $75^0$N and compares the microphysical properties (number concentration, effective diameter and ice water content) in these two "zones". Backward trajectories are used to identify the location and basically history of the air parcels sampled. Probably very difficult to do in practice, Lagrangian aircraft sampling along air trajectories would be a better way to identify the source regions and growth/evolution of the particles along the trajectories. Also, comparing southern and northern hemisphere cirrus within these latitudinal zones would be desirable.

I have numerous comments that the authors should consider prior to accepting this article for publication.

**Major Comments**

It's a pity that aerosol and trace gases, which were measured on the aircraft (line 162) were not used in the analysis. These measurements could have potentially provided insight into the resulting cirrus cloud properties. Also, ice nuclei and residual aerosols collected using a counterflow virtual impactor would have been very useful as part of the analysis. If you had measurements of black carbon, you could have identified contrail influenced cirrus.

At the beginning of Section 2, it is important to have a very clear and detailed statement about particle probe measurement accuracy and the reliability of the ice water content calculations when small particles dominate. Also, the CDP sample volume may be insufficient to collect sufficient particles in small sizes to measure low concentrations of very small particles.

Line 57. Liquid origin cirrus. Why do the ice particles need to ascend in the updrafts. Couldn't it be the commonly observed cloud top liquid water regions (from satellite lidar data, Zhang and others) that form ice crystals that grow and fall out?

Lines 153-155 and elsewhere. Could any of this cirrus been generated by deep convection? Is it possible that some of the cirrus is due to anvil from upstream convection?

Lines 237-238. Don't you have a liquid water probe or RICE probe? Relative humidity would also be sufficient to discern regions where liquid water was observed. Perhaps the temperatures are sufficient to rule out liquid water. Nonetheless, I feel that you are limiting your observations that could potentially result in misleading conclusions.

Line 242. I think that 10 sec is preferable. I suggest comparing 10 to 2 seconds. Later, you compare later measurements but it think all the comparisons should be done with 10-sec averages.

Lines 251-253. Could you determine whether commercial aircraft crossed the trajectories by using commercial aircraft tracks?

252, 262. How do you derive the IWC along backward trajectories. How about generating cells, snowbands, etc? How about generating cells, gravity waves, etc.

Figure 2. Add another panel that shows temperature would be very useful.

Table 2. IWCs below about 0.01 g/m3. How do you possibly derive accurately these very low IWCs. Is the mass dimensional relationship you cite reliable at the very small sizes associated with the low ice water contents?

Satellite-based radiometric measurements could be used to evaluate your findings.

Section 5. Very nice sensitivity studies

Minor Comments

I have a lot more minor comments but these are the main ones.

Line 1 significance>importance

9-11. How good are the small particle measurements-that could account for the differences you find in your analsysis.

20-21. How do you know this with certainty

22: scarce>relatively few

24-26. Important finding

45-47. In-situ generated cirrus form....

45-47. Cirrus can also form by cloud top radiative cooling without the need for lifting.

99-101. Concentration also decreases because the growing ice crystals have an increasing fall velocity.

188-190. The probe at minimum number counts would be 25/liter, which is often the concentration of ice in cirrus.

236. allows us to

348-350. Would this show up in back trajectories?

Figure 5 is really nice. It makes a strong case for your interpretations.

454-455. Not necessarily ascending motion.

Figure S3. Sensitivity rather than Sensibility.

Andy Heymsfield, NCAR

---

## Referee Comment (RC2)

Manuscript Review

Differences in microphysical properties of cirrus at high and mid-latitudes

Elena De La Torre Castro et al.

General comments

This study is a careful but detailed comparison between mid-latitude (ML) and high-latitude (HL) cirrus by evaluating the differences in three, microphysical bulk properties of these high altitude clouds – number concentration, median effective diameter and ice (liquid) water content . I commend the authors on the clarity of their presentation and the plausible explanations of why there are significant differences in the microphysical properties. I also believe that this is the first refereed publication on cirrus that has explicitly stratified the results according to the latitude of cloud formation and the latitude of cloud measurements. This additional analysis step has led to a better understanding of cirrus formation and evolution and will, no doubt, be used in future studies to better label cirrus clouds.

I do have three major concerns that I would like the authors to address, as I think that there are possibly two environmental factors tha could alter some of their conclusions, or at least open the discussion that could further explain some of the differences that are observed.

My first concern is related to where the clouds were measured with respect to the tropopause. The authors mention that the upper tropopause, in both the ML and HL, were frequently supersaturated but it appears that the analysis has not stratified the cirrus with respect to their altitude with respect to the location of the tropopause, nor is it stated whether any of these clouds were actually in the stratosphere. Given that it appears that there were either no aerosol measurements on the project, or that they are not being evaluated within the context of cloud formation, and given that many times stratospheric and tropospheric aerosol particles can be quite different in their composition, and hence activity as CCN or IN, I strongly recommend that this study add this factor into the analysis. Is it possible that some of these cirrus are polar stratospheric clouds?  These are more common in the winter but cannot be completely ruled out in the summer and their formation at high latitudes typical are on a different type of aerosol than in the troposphere,

My second concern is related to more general synoptic features that not only are important in how they can lead to vertical motion, but also how they can lead to horizontal and vertical shear. Very little is said about mesoscale and synoptic scale features like the polar vortex that is weaker in summer than winter but can play a major role in determining the history of the air masses where cirrus are found.

My third concern is that it appears that a Cloud and Aerosol Spectrometer with Polarization Detection (CAS-POL) was also a part of the instrument complement but measurements of the polarization ratio of small ice crystals were not used. A great deal more information about the small crystal tail (Fig. 8) could be extracted by comparing the relative shapes of crystals < 50 μm in the four cirrus types shown in this figure. Rather than just speculating about possible contrail crystals, there is a high probability that this could be addressed quantitively.

Other comments, questions and suggestions.

Line 154: How are convective cirrus identified?  Even though the paper says they are excluded from the analysis, at several points in the paper it appears that they have been included in at least some of the analysys (e.g. lines 344 and 487).

Line 167: Here and elsewhere, DMT Inc. should be changed to DMT LLC.

Line 175: Why is the CDP data being truncated at 37.5 um?

Line 199&200: The definitions for the shadowing levels are a bit nebulous. Is 0-25% being classified as unshadowed? Also, the importance of the grayscale is more because it helps constrain the depth of field and not so much shape analysis.

Line 207: Although the PADS data system uses PAS (Particle Air Speed) to denote the pitot measurement, it measures air speed not particle speed, so this should be changed in the manuscript from PAS to TAS.

Line 222,223: I am assuming that the 50μm lower threshold stated here is assuming that when only ½ a dide is shadowed it will be registered as "on", but probably using 100 μm as the lower threshold is a better estimate.

Line 292: "higher sample efficiencies" I think is meant to say "higher sample volumes".

Line 295: I do not agree with this statement that "maximum dimension diameter represents more accurately the spatial extent of ice crystals, which is key for radiative impact calculations." It is the cross-sectional area of crystals that is key to radiative impact calculations. To be sure, our measurement community has not reached a consensus on best dimensions to use in calculating various quantities but unless the authors can show that the end result of using maximum diameter as opposed to area equivalent diameter is marginally different, than I think that this justification should, at the least, be removed.

Line 362: For consistency with the other three categories, change "cirrus measured and formed at mid-latitudes" to "cirrus formed and measured at mid-latitudes".

Figure 8: Strongly recommend showing extinction or area vs size to highlight the radiative impact, on a linear scale. This would show even more clearly the differences in cirrus properties and since the impact on radiative forcing is mentioned numerous times, why not display the data in a way that emphasizes these differences?

This is just a suggestion, but I would ask the authors to consider looking at the particle by particle data from the imaging probes, plot the interarrival times and compare regions where they think there might be contrail cirrus with regions of similar concentration but where contrail like are not present. I think that this might highlight clusters of small contrail crystals embedded in the natural cirrus. Just a thought.

---

## Author Comment (AC1)

We would like to thank both reviewers for their constructive comments and helpful suggestions. They led to interesting discussions and to a higher quality of the revised manuscript. We made corrections, included further analyses suggested by the reviewers and incorporated 3 figures in the main text and supplement, as well as paragraphs referring to the new analyses in the main text. All the responses (in blue) and corresponding questions (in black) of the reviewers are listed below. Copied text from the manuscript is marked in italics and underlined sentences correspond to changes in the manuscript (see also the submitted manuscript with track-changes). We also changed the color map of the figures 2, 4, 5 and 7 since the rainbow color map is not recommended.

**Questions and responses to Reviewer 1**

Review of "Differences in microphysical properties of cirrus at high and mid-latitudes", EGUsphere 2023-374, by Castro, Jurkat-Witschas, Afchine, Grewe, Hahn, Kirschler, Kramer, Lucke, Spelten, Wernli, Zoger, and Voigt.

This is an interesting and novel study that uses data from the CIRRUS-HL field program that collected airborne observations at latitudes from about 35° to 60°N and 60° to about 75°N and compares the microphysical properties (number concentration, effective diameter and ice water content) in these two "zones". Backward trajectories are used to identify the location and basically history of the air parcels sampled. Probably very difficult to do in practice, Lagrangian aircraft sampling along air trajectories would be a better way to identify the source regions and growth/evolution of the particles along the trajectories. Also, comparing southern and northern hemisphere cirrus within these latitudinal zones would be desirable.

We thank Andrew Heymsfield for rating this study as bearer of new and interesting information. We agree that a Lagrangian aircraft sampling would be a better way to study the evolution of the cirrus properties along their life-cycle and it is an idea that could be considered in future campaigns. However, as the reviewer indicates, it would be very difficult to do it in practice or at least in a sufficiently large number of cases to enable a statistical analysis, as in this study. The last aspect is also very interesting and we plan indeed to perform a similar analysis in the southern hemisphere with the next HALO campaign in 2025 (HALO-South). A comparison with the northern hemisphere cirrus microphysical properties from the present study will be addressed then.

I have numerous comments that the authors should consider prior to accepting this article for publication.

Major
Comments

It's a pity that aerosol and trace gases, which were measured on the aircraft (line 162) were not used in the analysis. These measurements could have potentially provided insight into the resulting cirrus cloud properties. Also, ice nuclei and residual aerosols collected using a counterflow virtual impactor would have been very useful as part of the analysis. If you had measurements of black carbon, you could have identified contrail influenced cirrus.

We thank the reviewer for this thoughtful comment. The platform that we used (HALO aircraft) allows a wide variety of instrumentation, which is definitely very profitable. There will be an overview publication on the different data, methods and results from this campaign (Voigt et al. in prep.). However, we cannot cover all these different instruments, methods and topics in the current study. Here, we focus on the cirrus microphysical properties from three cloud probes, since the aerosol and trace gases colleagues are also analyzing their data and publications on that matter are being prepared. A follow up to this publication will focus on a further analysis including aerosol data from experiments and a numerical model to address the aerosol load and type along the trajectory and also compare the model with in-situ measurements of aerosol.

At the beginning of Section 2, it is important to have a very clear and detailed statement about particle probe measurement accuracy and the reliability of the ice water content calculations when small particles dominate. Also, the CDP sample volume may be insufficient to collect sufficient particles in small sizes to measure low concentrations of very small particles.

We thank the reviewer for this comment and we believe that we already addressed these issues in the manuscript. A general statement on the particle size and concentration measurements is given in Section 3.1 in the lines 223-225:

*Baumgardner et al. (2017) estimated the error in sizing in ±20% for ice particles of diameters larger than 100 − 200 μm and the error in concentration in ±50%. For smaller particles, the error is larger and increases inversely proportional to the diameter.*

We believe that Section 3.1 is more appropriate to describe this because it comprises all the information regarding the instruments and methods applied and Section 2 is rather thought as an overview of the campaign and therefore contains more general information. We also comment on the lower limit of the CDP due to its smaller sample volume compared to the optical array probes (CIP and PIP), in lines 187-191:

*The effects of the limited sampling volumes in the scattering probes were extensively explained by Krämer et al. (2020) in the appendix A2.3 with focus on the CAS-DPOL. In our case, the CDP has a lower N limit ~ 0.025 $cm^{-3}$, when only one particle is recorded in one second. These so-called "single-particle-events" cause an increase in the frequency of the low N range of the CDP. We mitigated this effect by eliminating the single-particle-events due to their low statistics and inaccuracy.*

In general, we do not find many situations where the small particles dominate (except for contrail crossings). An overestimation of N and IWC would take place in sequences of shattering, however we did not find evidence for shattering in the CDP and therefore, we do not expect the contribution of the small particles to the total IWC to be substantial.

Line 57. Liquid origin cirrus. Why do the ice particles need to ascend in the updrafts. Couldn't it be the commonly observed cloud top liquid water regions (from satellite lidar data, Zhang and others) that form ice crystals that grow and fall out?

We thank the reviewer for this comment but we believe that maybe our statement was misleading. With the ascent reference in the sentence: "*Liquid origin cirrus form from mixed-phase clouds at lower altitudes that glaciate as they ascend and rise to the cirrus heights*" we refer to the fact that some cirrus are formed at lower altitudes as mixed-phase clouds and to the need for their ascent to higher altitudes and their complete glaciation to be considered "cirrus". In our measurements, we look at cirrus samples below -38°C, and therefore, when looking at their origin, if they had a liquid origin, they had to ascend to the cirrus altitudes. If there are any remaining liquid droplets, they freeze when the temperature descends below -38 °C. For this reason, we think that the definition of the liquid origin cirrus in line 57 is appropriate.

Lines 153-155 and elsewhere. Could any of this cirrus been generated by deep convection? Is it possible that some of the cirrus is due to anvil from upstream convection?

Embedded convection that occurred as part of the ice cloud formation driven by a warm conveyor belt cannot be disentangled from low updraft cirrus and might not be captured by the backward trajectories. However, except for two dedicated flights (F12 and F15) that are excluded from the analysis for that reason, we avoided convectively formed cirrus. If the reviewer wonders whether also some of the measurements originated from the anvil of upstream convection, we confirm that we identified the presence of convective systems in some of our transfer flights home from the target region. We identified them with the help of the flight reports and ECMWF forecast for the flight planning. Therefore, it is possible that some isolated cirrus measurement in these cases were generated by deep convection. In those cases, we observed that they were characterized by large ED and relatively high number concentration. However, these events were rare and not targeted. We included the following sentence in line 154 to provide a more explicit clarification on this issue:

*However, during some return transfers, outflows of single convective systems were encountered over Germany. These isolated events have not been excluded, as they are inseparable from the surrounding cirrus deck.*

Lines 237-238. Don't you have a liquid water probe or RICE probe? Relative humidity would also be sufficient to discern regions where liquid water was observed. Perhaps the temperatures are sufficient to rule out liquid water. Nonetheless, I feel that you are limiting your observations that could potentially result in misleading conclusions.

Unfortunately, the Hotwire Liquid Water Content Sensor attached to the CAS did not work and we did not have any further liquid water probe on board. Therefore, we define the temperature threshold of -38°C in order to rule out liquid water and ensure that we are only considering ice clouds. There could be samples above -38°C with only ice that could be included in our analysis and it is true that we are limiting our observations but, in some way, including warmer samples that potentially contain

liquid water would be more misleading and ambiguous. In that sense, we are just looking at cirrus clouds below -38°C and drawing conclusions about those measurements noting that these cirrus might be a subsample of the full ice cloud regime. We take the reviewer's concern into account and modify the lines 237-238 in the following way:

*Even though ice clouds can exist at higher temperatures, we consider only measurements below -38° C to ensure completely glaciated clouds, and calculate the N, ED, and IWC. This might exclude some cirrus measurements above the threshold, but guarantees that no liquid water is present.*

Line 242. I think that 10 sec is preferable. I suggest comparing 10 to 2 seconds. Later, you compare later measurements but I think all the comparisons should be done with 10-sec averages.

For the derivation of the cirrus properties we find it more appropriate to 2-sec averages for the HALO aircraft measurements at high velocities, since 10 sec corresponds to horizontal distances of ~2 km in our case, where the properties might be quite different or it could even include a no cloud sequence, suppressing the effect of local inhomogeneities. Krämer et al., (2020) used a time resolution with the minimum possible resolution of 1 s for representing the cirrus climatology precisely to ensure a sufficient spatial resolution of the cirrus. Patnaude et al., (2021) also used a 1-sec time resolution for their climatology. We find a time resolution of 2 sec as a good compromise, as it also makes our work more comparable to the mentioned studies.

Lines 251-253. Could you determine whether commercial aircraft crossed the trajectories by using commercial aircraft tracks?

In general, we were operating in dense air traffic regions over Central Europe and that is why we expected an influence of air traffic in our measurements. During the campaign we could directly identify a few contrail crossings to the corresponding aircraft and the information was incorporated in the flight reports. However, this topic is being investigated separately and will be addressed in another publication.

252, 262. How do you derive the IWC along backward trajectories. How about generating cells, snowbands, etc? How about generating cells, gravity waves, etc.

IWC was interpolated in the same way as other variables (e.g., temperature) from the gridded ECMWF analysis fields to the position of the trajectories. Therefore, our IWC values are a direct product of the ECMWF data assimilation procedure. Details on the model can be found in the ECMWF documentation in the references. The data have a limited spatial resolution of about 15 km (but we use the data on a slightly coarser grid) and therefore do not capture, e.g., single convective cells. However, several comparisons of in situ observations of ice clouds and ECMWF products have shown that the latter capture most of the major features in the upper troposphere (Wernli et al., 2016). Also, we would not know of a higher-resolution product of IWC, which would be available for a domain as large as the one covered by this campaign.

Figure 2. Add another panel that shows temperature would be very useful.

We thank the reviewer for this suggestion but since we show already in Fig. 3 the temperature ranges and the temperature distribution for high and mid latitudes, it might not be necessary to include it here. In addition, a similar graph to N, ED and IWC latitude dependence might not be appropriate for temperature, since large distances were covered at a certain flight level and therefore, under the same temperature.

Table 2. IWCs below about 0.01 g/m3. How do you possibly derive accurately these very low IWCs. Is the mass dimensional relationship you cite reliable at the very small sizes associated with the low ice water contents?

It is true that there are several concerns associated to the mass-dimension relationship we used (from Heymsfield et al., (2010)) regarding small particles. However, according to the cited article, it appears to be a problem when the IWC is low and small particles dominate (due to shattering, homogeneous nucleation, strong updrafts…). In our case, the low IWCs are mainly produced by the bigger but few ice crystals (from the PIP) and not particularly from the small particles. In the following graph, where IWC is represented against ED, the red rectangle indicates the region of low IWC of small particles (range of the CDP) and we do not observe many data points in this area, whereas more frequently low IWCs are associated to larger EDs (larger ice particles from the CIPg and PIP contributions). Here, the irregularity of the large ice crystals, which differ from spheres, would be the largest source of uncertainty. The difference between the use of different m-D relationships was assessed by Afchine et al., (2018) who found out that the resulting IWCs differed by a factor of 1.5 at most.

[Figure]

Satellite-based radiometric measurements could be used to evaluate your findings.

Thank you for this suggestion, we also think it would be very interesting but it is rather out of scope for this study. This wide dataset will for sure enable comparison with satellite retrievals and we will encourage this cooperation to address the topic in further publications, similar to Wang et al., (2023).

Section 5. Very nice sensitivity studies

We are very pleased that the reviewer gave a positive assessment of the sensitivity studies.

Minor
Comments

I have a lot more minor comments but these are the main ones.

Line 1 significance>importance

We thank the reviewer for noting this and we changed it in the manuscript.

9-11. How good are the small particle measurements-that could account for the differences you find in your analsysis.

The differences that we find are mainly driven by the data in the CIP and not by the small particles in the CDP, as already commented. We do not find shattering in the CDP from the interarrival time analysis, that would lead to artificially increased number concentrations. In general, we observed low particle number concentrations in the CDP. Therefore, the main differences are not expected to be controlled by the smaller particles.

20-21. How do you know this with certainty

We make this statement based on the analyses of sections 4.2 and 4.3. The classification of liquid and in situ origin is conducted looking at the IWC and LWC along the backward trajectories. We could then identify the fraction of cirrus of in situ origin and of liquid origin that are found in the three cirrus groups defined according to the latitude. It is not meant as a general statement and the properties that we find are restricted to the measurements during this campaign. Further campaigns could help to extend our observation of the cirrus properties according to this latitude classification.

22: scarce>relatively few

We adopt the recommendation of the reviewer and changed this in the manuscript.

24-26. Important finding

We thank the reviewer for acknowledging the relevant finding of this study

45-47. In-situ generated cirrus form....

We thank the reviewer for this note and we modified the paragraph to also include the nucleation processes of liquid origin cirrus (homogeneous nucleation of the liquid droplets or heterogeneous nucleation on INPs), since we want to make a general

statement here to introduce the nucleation mechanisms of cirrus clouds.

*Cirrus are generally formed by air cooling from rising air masses or by radiative cooling, either by homogeneous nucleation of soluble aerosol particles of water droplets, or heterogeneous nucleation of insoluble ice nucleating particles (INPs) (Kärcher and Lohmann, 2002; Hoose and Möhler, 2012). Homogeneous nucleation of soluble aerosol particles occurs most frequently at strong cooling rates and high supersaturations with respect to ice (RHi ~ 150%).*

45-47. Cirrus can also form by cloud top radiative cooling without the need for lifting.

We modified the paragraph to make it clear that there are other formation processes, to take into account the reviewer's consideration.

*Cirrus are generally formed by air cooling from rising air masses or by radiative cooling, either by homogeneous nucleation of soluble aerosol particles or water droplets, or heterogeneous nucleation of insoluble ice nucleating particles (INPs) (Kärcher and Lohmann, 2002; Hoose and Möhler, 2012).*

99-101. Concentration also decreases because the growing ice crystals have an increasing fall velocity.

We thank the reviewer for this remark and included the following sentence:

*During the transition from contrail to cirrus, the ice crystals grow by uptake of water vapor and the ice number concentration decreases due to mixing with ambient dry air and the increasing fall velocity of the ice crystals (Schröder et al., 2000; Unterstrasser and Gierens, 2010; Kübbeler et al., 2011; Voigt et al., 2017; Schumann et al., 2017; Grewe et al., 2017).*

188-190. The probe at minimum number counts would be 25/liter, which is often the concentration of ice in cirrus.

We are not sure what the reviewer is indicating. We agree that the CDP has that lower limit and we explain in those lines that we exclude single-particle events for that reason. The CIPg and PIP have larger sample volumes, with lower limits of approximately 0.052 counts/L and 0.003 counts/L, respectively, and therefore can provide concentration in cirrus lower than the usual concentration of ice in cirrus and thus represent the cirrus properties with more accuracy.

236. allows us to

Thanks for pointing to the spelling mistake, we changed it accordingly in the revised manuscript.

348-350. Would this show up in back trajectories?

The topic of identifying contrails from the flight tracks and backward trajectories is also discussed in our reply to the question about lines 251-253 from Reviewer #2. We refer here again to the future publication in preparation on the contrails during CIRRUS-HL.

However, we also considered applying other methods, which were applied in previous publications (Li et al., 2022), where contrail cirrus candidates were selected whether they fulfilled the Schmidt-Appleman criterion or not. However, since the focus of this work was not on aviation effects, we focused on indicating possible connections between events. Here, from the information collected in the flight reports, weather forecasts and the updraft velocities, we found a strong indication that those high N and low ED were caused by contrails and contrail cirrus.

Figure 5 is really nice. It makes a strong case for your interpretations.

We thank the reviewer for this nice comment.

454-455. Not necessarily ascending motion.

We modified the sentence to account for other processes in the following way:

*Ascending motion, isobaric mixing or radiative cooling can reduce the temperature in the cloud with a subsequent increase in relative humidity.*

Figure S3. Sensitivity rather than Sensibility.

We thank the reviewer for noting this and we changed it in the manuscript.

Andy Heymsfield, NCAR

**References**

Afchine, A., Rolf, C., Costa, A., Spelten, N., Riese, M., Buchholz, B., Ebert, V., Heller, R., Kaufmann, S., Minikin, A., Voigt, C., Zöger, M., Smith, J., Lawson, P., Lykov, A., Khaykin, S., and Krämer, M.: Ice particle sampling from aircraft – influence of the probing position on the ice water content, Atmos. Meas. Tech., 11, 4015–4031, https://doi.org/10.5194/amt-11-4015-2018, 2018.

ECMWF: IFS Documentation CY47R1 - Part IV: Physical Processes, 4, ECMWF, https://doi.org/10.21957/cpmkqvhja, 2020

Heymsfield, A. J., C. Schmitt, A. Bansemer, and C. H. Twohy, 2010: Improved Representation of Ice Particle Masses Based on Observations in Natural Clouds. J. Atmos. Sci., 67, 3303–3318, https://doi.org/10.1175/2010JAS3507.1.

Krämer, M., Rolf, C., Spelten, N., Afchine, A., Fahey, D., Jensen, E., Khaykin, S., Kuhn, T., Lawson, P., Lykov, A., Pan, L. L., Riese, M., Rollins, A., Stroh, F., Thornberry, T., Wolf, V., Woods, S., Spichtinger, P., Quaas, J., and Sourdeval, O.: A microphysics guide to cirrus – Part 2: Climatologies of clouds and humidity from observations, Atmospheric Chemistry and Physics, 20, 12 569–12 608, https://doi.org/10.5194/acp-20-12569-2020, 2020.

Li, Y., Mahnke, C., Rohs, S., Bundke, U., Spelten, N., Dekoutsidis, G., Groß, S., Voigt, C., Schumann, U., Petzold, A., and Krämer, M.: Upper tropospheric slightly ice-subsaturated regions: Frequency of occurrence and statistical evidence for the

appearance of contrail cirrus, Atmospheric Chemistry and Physics Discussions, 2022, 1–32, https://doi.org/10.5194/acp-2022-632, 2022.

Patnaude, R., Diao, M., Liu, X., and Chu, S.: Effects of thermodynamics, dynamics and aerosols on cirrus clouds based on in situ observations and NCAR CAM6, Atmos. Chem. Phys., 21, 1835–1859, https://doi.org/10.5194/acp-21-1835-2021, 2021.

Unterstrasser, S. and Gierens, K.: Numerical simulations of contrail-to-cirrus transition – Part 2: Impact of initial ice crystal number, radiation, stratification, secondary nucleation and layer depth, Atmospheric Chemistry and Physics, 10, 2037–2051, https://doi.org/10.5194/acp-10-2037-2010, publisher: Copernicus GmbH, 2010.

Wang, Z., Bugliaro, L., Jurkat-Witschas, T., Heller, R., Burkhardt, U., Ziereis, H., Dekoutsidis, G., Wirth, M., Groß, S., Kirschler, S., Kaufmann, S., and Voigt, C.: Observations of microphysical properties and radiative effects of a contrail cirrus outbreak over the North Atlantic, Atmos. Chem. Phys., 23, 1941–1961, https://doi.org/10.5194/acp-23-1941-2023, 2023.

Wernli, H., M. Boettcher, H. Joos, A. K. Miltenberger, and P. Spichtinger (2016), A trajectory-based classification of ERA-Interim ice clouds in the region of the North Atlantic storm track, Geophys. Res. Lett., 43, 6657–6664, doi:10.1002/2016GL068922.

**Questions and answers to Reviewer 2**

**General comments**

This study is a careful but detailed comparison between mid-latitude (ML) and high-latitude (HL) cirrus by evaluating the differences in three, microphysical bulk properties of these high altitude clouds – number concentration, median effective diameter and ice (liquid) water content . I commend the authors on the clarity of their presentation and the plausible explanations of why there are significant differences in the microphysical properties. I also believe that this is the first refereed publication on cirrus that has explicitly stratified the results according to the latitude of cloud formation and the latitude of cloud measurements. This additional analysis step has led to a better understanding of cirrus formation and evolution and will, no doubt, be used in future studies to better label cirrus clouds.

We are very grateful to Darrel Baumgardner for highlighting the importance and relevance of this study and for his positive feedback.

I do have three major concerns that I would like the authors to address, as I think that there are possibly two environmental factors that could alter some of their conclusions, or at least open the discussion that could further explain some of the differences that are observed.

My first concern is related to where the clouds were measured with respect to the

tropopause. The authors mention that the upper tropopause, in both the ML and HL, were frequently supersaturated but it appears that the analysis has not stratified the cirrus with respect to their altitude with respect to the location of the tropopause, nor is it stated whether any of these clouds were actually in the stratosphere. Given that it appears that there were either no aerosol measurements on the project, or that they are not being evaluated within the context of cloud formation, and given that many times stratospheric and tropospheric aerosol particles can be quite different in their composition, and hence activity as CCN or IN, I strongly recommend that this study add this factor into the analysis. Is it possible that some of these cirrus are polar stratospheric clouds? These are more common in the winter but cannot be completely ruled out in the summer and their formation at high latitudes typical are on a different type of aerosol than in the troposphere,

We thank the reviewer for raising this concern and we agree that this factor should be included in the analysis. Therefore, we calculated the altitude of the cirrus with respect to the tropopause height and included in the supplement an absolute frequency distribution graph of the cirrus altitudes with respect to the tropopause (tropopause data sets for the CIRRUS-HL period from ERA5 reanalysis data available in https://datapub.fz-juelich.de/slcs/tropopause/data/v2/era5/2021/, Hoffmann and Spang, 2022). From the presented cirrus data set, 2.6% of the data points were above the tropopause (2.4% at mid latitudes and 0.2% at high latitudes). With this added information, we can rule out the possibility of having measured polar stratospheric clouds. Additionally, the distribution of cirrus altitudes with respect to the tropopause is similar for both high and mid-latitude cirrus groups. Therefore, we conclude that the cirrus in our data set were almost entirely measured in the upper troposphere and very few measurements in the stratosphere. We find that this analysis provides more information on the cirrus formation processes and included a paragraph at the end of Section 4.1, after line 357:

*Furthermore, we investigated whether some of the cirrus were measured in the stratosphere, since extratropical convection penetrating the stratosphere is most likely to occur during summer (Dessler, 2009). In addition, the height of the tropopause (TP) varies strongly between 35 and 50° N, and cirrus measurements at high and mid latitudes may be distributed differently with respect to the height of the TP (Dessler, 2009). In order to derive the altitude of the cirrus measurements with respect to the TP, the TP height was extracted from the Hoffmann and Spang (2021) dataset and the definition used was the WMO (World Meteorological Organization) 1st TP height from the ECMWF reanalysis ERA5 data (Hoffmann and Spang, 2022). The absolute frequency distribution of the HL and ML cirrus altitudes with respect to the TP is shown in Fig. S2 and reveals similar profiles between mid and high latitudes. From all data points, 2.6% were measurements above the TP, with only 0.2% belonging to HL cirrus and most of the measurements were performed in the upper troposphere at about 1 km below the TP. We do not observe a substantial influence of stratospheric clouds, with an even smaller contribution at high latitudes.*

My second concern is related to more general synoptic features that not only are

important in how they can lead to vertical motion, but also how they can lead to horizontal and vertical shear. Very little is said about mesoscale and synoptic scale features like the polar vortex that is weaker in summer than winter but can play a major role in determining the history of the air masses where cirrus are found.

We fully agree, for understanding specific episodes or case studies, such a detailed analysis would be essential, but our main interest in this study is the statistical comparison of high vs. mid-latitude ice cloud properties. Therefore, we take into account the reviewer's concern and provide a brief summary of the general synoptic evolution during the campaign in Section 2 after line 147:
*Over Western Europe, the unusually frequent occurrence of upper-level troughs and cutoff lows led to statically unstable situations with many thunderstorms and hail over Western and Central Europe. Over the eastern North Atlantic, several extratropical cyclones with warm conveyor belts occurred, in particular on 7 and 12 July 2021 (according to ECMWF weather forecasts and flight reports).*

My third concern is that it appears that a Cloud and Aerosol Spectrometer with Polarization Detection (CAS-POL) was also a part of the instrument complement but measurements of the polarization ratio of small ice crystals were not used. A great deal more information about the small crystal tail (Fig. 8) could be extracted by comparing the relative shapes of crystals < 50 μm in the four cirrus types shown in this figure. Rather than just speculating about possible contrail crystals, there is a high probability that this could be addressed quantitively.

We thank the reviewer for this comment and agree that the depolarization ratio information from the CAS-DPOL could contribute to identify more efficiently contrail particles. Even though we also performed measurements with a CAS-DPOL in this campaign, we did not have the option for this study to use the depolarization ratios from this instrument, as the work on the CAS-DPOL depolarization ratio calibration is ongoing and this will be used for contrail analysis and incorporated in another publication. Therefore, we could not include these measurements in the present study. We decided to use the CDP-CIPG-PIP combination for cirrus properties and the CAS-DPOL for contrails and contrail cirrus properties.

Other comments, questions and suggestions.

Line 154: How are convective cirrus identified? Even though the paper says they are excluded from the analysis, at several points in the paper it appears that they have been included in at least some of the analysis (e.g. lines 344 and 487).

We thank the reviewer for this comment and we agree that this might have been a bit confusing in the manuscript. F12 and F15 are classified as "convective flights" in Section 2 due to the targets defined for those flights. Only in these two flights, we targeted convective systems nearby and over the mountain regions of the Alps. Since during these flights we did not measure other type of cirrus, we decided to exclude these flights of our analysis because the focus was not set on convection. Regarding the lines 344 and 487, where we use the words "isolated convective systems", we

refer to isolated encounters of outflows of convective systems that were measured over Germany during the transfer back from the target measurement regions. We discovered these outliers during the analysis of the data and we found no reason to exclude these isolated data points, since they were identifiable and explainable, since the higher N and larger EDs found in these cases were connected to reported convective systems from the flight reports. In addition, for the sake of transparency we rather analyze all cirrus data points within a flight and remove flights whose targets are not covered in this study. For the reader's clarity, we included a sentence in line 154: […] and have been excluded from the present analysis since convection is not an objective of this study. *However, during some return transfers, outflows of single convective systems were encountered over Germany. These isolated events have not been excluded, as they are inseparable from the surrounding cirrus deck.*

Line 167: Here and elsewhere, DMT Inc. should be changed to DMT LLC.

We thank the reviewer for noting this and we included the correction in the revised manuscript.

Line 175: Why is the CDP data being truncated at 37.5 um?

Since the CDP and CIP particle size ranges overlap between 15 and 50 µm, we need to define where to change from using CDP to CIP data. We set our limit on 37.5 µm because it corresponds to already 3 pixels on the CIP (approximately a 3x3 pixel image) which can already reflect particle shapes and provide reliable size values. It is preferable to use the CIP at the lowest possible limit to improve statistics due to the larger sample volume of the CIP compared to the CDP for sizes above 30 µm. We find this value as a comprise between using the CIP before the CDP but making sure that the CIP images are large enough to provide reasonable data.

Line 199&200: The definitions for the shadowing levels are a bit nebulous. Is 0-25% being classified as unshadowed? Also, the importance of the grayscale is more because it helps constrain the depth of field and not so much shape analysis.

We agree on this remark and reformulated those sentences in the following form:

*On the contrary, the CIPgs is a grayscale probe and includes two more shadow intensity levels (25, 50, and 75%, nominal thresholds) that define four pixel states: no shadow, at least 25, 50, and 75% obscured. This functionality allows to record images already at 25% dimming, which helps to constrain the depth of field (DoF) and adds details for the particle shape analysis.*

Line 207: Although the PADS data system uses PAS (Particle Air Speed) to denote the pitot measurement, it measures air speed not particle speed, so this should be changed in the manuscript from PAS to TAS.

We thank the reviewer for this suggestion but we refer to PAS in the manuscript as Probe Air Speed and not Particle Air Speed (introduced in line 177) and we would

prefer to stick with PAS in order to differentiate the air speed measured by the probes with the one measured by the aircraft meteorological data system BAHAMAS.

Line 222,223: I am assuming that the 50μm lower threshold stated here is assuming that when only ½ a dide is shadowed it will be registered as "on", but probably using 100 μm as the lower threshold is a better estimate.

We agree on this observation and changed it accordingly.

Line 292: "higher sample efficiencies" I think is meant to say "higher sample volumes".

Yes, thank you! We corrected it in the manuscript.

Line 295: I do not agree with this statement that "maximum dimension diameter represents more accurately the spatial extent of ice crystals, which is key for radiative impact calculations." It is the cross-sectional area of crystals that is key to radiative impact calculations. To be sure, our measurement community has not reached a consensus on best dimensions to use in calculating various quantities but unless the authors can show that the end result of using maximum diameter as opposed to area equivalent diameter is marginally different, than I think that this justification should, at the least, be removed.

We thank the reviewer for this comment and he is completely right, we do not have a proof for this statement, it is just a believe or intuition that should not be stated here and therefore we removed it from the manuscript.

Line 362: For consistency with the other three categories, change "cirrus measured and formed at mid-latitudes" to "cirrus formed and measured at mid-latitudes".

Thanks for catching this! We have changed it in the manuscript.

Figure 8: Strongly recommend showing extinction or area vs size to highlight the radiative impact, on a linear scale. This would show even more clearly the differences in cirrus properties and since the impact on radiative forcing is mentioned numerous times, why not display the data in a way that emphasizes these differences?

We thank the reviewer for this recommendation and we totally agree that the extinction should be shown in order to reflect better the radiation impact of the cirrus measured during this campaign. We have taken the suggestion and we included two figures on the cirrus extinction in the main text (Fig. 8(c)) and in the supplement (Fig. S5). We calculated the extinction using the PSD of the CDP and the images surface from the particle-by-particle data from the CIP and PIP (Schumann et al., 2011; Mitchell et al., 2011; Thornberry et al., 2017). We show the relative frequency distribution of the extinction of the four cirrus types (in situ origin ML cirrus, liquid origin ML cirrus, in situ origin HL cirrus, and liquid origin HL cirrus) in Fig. 8(c) and also

include an overview of the relative frequency of the extinction as a function of latitude (as in Fig. 2) in Fig. S5. We mention this topic in the main text by including the following sentences:

After line 241:
*We derive the extinction coefficient (βext) from the projected area of the CIPg and PIP particle images and spheres with the arithmetic mean diameters of the CDP particle size distribution (PSD) (Schumann et al., 2011; Mitchell et al., 2011). The extinction efficiency Qext is approximated to 2 assuming short wavelengths relative to the particles' dimension (Thornberry et al., 2017).*

After line 279:

*A similar behavior is observed in the normalized frequency distribution of βext, which is shown in Fig. S5 of the Supplement.*

After line 494:

*In Fig. 8(c) we show another way to represent the radiative impact of cirrus through βext, which is often used to assess the climate impact of contrails (Bräuer et al., 2021a; Voigt et al., 2021). For a certain IWC, the extinction is larger when the ice mass is distributed in smaller crystals (Unterstrasser and Gierens, 2010). We observe higher extinction coefficients for the liquid origin cirrus at HL and ML, with slightly higher median for ML, compared to the in situ origin cirrus. This is expected due to the higher IWC of the liquid origin cirrus. The lowest extinction corresponds to the in situ origin HL cirrus, since the N is lower and the ice particles are larger, which reduces the extinction and the radiative impact. In line with the PSDs, here, we also see a stronger difference between the in situ cirrus at HL and ML with a shift to larger extinctions at ML. In addition, both frequency distributions of ML cirrus show a small mode at 1 km−1, which is frequently observed in young contrails (Febvre et al., 2009; Voigt et al., 2011; Gayet et al., 2012; Bräuer et al., 2021a).*

This is just a suggestion, but I would ask the authors to consider looking at the particle by particle data from the imaging probes, plot the interarrival times and compare regions where they think there might be contrail cirrus with regions of similar concentration but where contrail like are not present. I think that this might highlight clusters of small contrail crystals embedded in the natural cirrus. Just a thought.

We find this suggestion very interesting and we did not think about that before so we thank the reviewer for this observation. We have had a look at the interarrival times of the particle by particle data from the CIP for the flight F17, where contrail cirrus was predicted and reported during the flight, and compared the sequences with natural cirrus but we could not identify a clear indication from the interarrival time analysis. On the contrary, younger contrails were easier to identify from the concentration and

IAT in the CDP, since natural cirrus measurements are not dominated by small particles present in the CDP.

**References**

Bräuer, T., Voigt, C., Sauer, D., Kaufmann, S., Hahn, V., Scheibe, M., Schlager, H., Diskin, G. S., Nowak, J. B., DiGangi, J. P., Huber, F., Moore, R. H., and Anderson, B. E.: Airborne Measurements of Contrail Ice Properties—Dependence on Temperature and Humidity, Geophysical Research Letters, 48, e2020GL092 166, https://doi.org/10.1029/2020GL092166, e2020GL092166 2020GL092166, 2021a.

Dessler, A. E.: Clouds and water vapor in the Northern Hemisphere summertime stratosphere, Journal of Geophysical Research: Atmospheres, 114, https://doi.org/10.1029/2009JD012075, 2009.

Febvre, G., Gayet, J.-F., Minikin, A., Schlager, H., Shcherbakov, V., Jourdan, O., Busen, R., Fiebig, M., Kärcher, B., and Schumann, U. (2009), On optical and microphysical characteristics of contrails and cirrus, J. Geophys. Res., 114, D02204, doi:10.1029/2008JD010184.

Gayet, J.-F., Shcherbakov, V., Voigt, C., Schumann, U., Schäuble, D., Jessberger, P., Petzold, A., Minikin, A., Schlager, H., Dubovik, O., and Lapyonok, T.: The evolution of microphysical and optical properties of an A380 contrail in the vortex phase, Atmospheric Chemistry and Physics, 12, 6629–6643, https://doi.org/10.5194/acp-12-6629-2012, 2012.

Hoffmann, L. and Spang, R.: An assessment of tropopause characteristics of the ERA5 and ERA-Interim meteorological reanalyses, Atmos. Chem. Phys., 22, 4019–4046, https://doi.org/10.5194/acp-22-4019-2022, 2022.

Hoffmann, L. and Spang, R.: Reanalysis Tropopause Data Repository, https://doi.org/10.26165/JUELICH-DATA/UBNGI2, 2021.

Mitchell, D. L., Lawson, R. P., and Baker, B.: Understanding effective diameter and its application to terrestrial radiation in ice clouds, Atmospheric Chemistry and Physics, 11, 3417-3429, https://doi.org/10.5194/acp-11-3417-2011, publisher: Copernicus GmbH, 2011.

Schumann, U., Mayer, B., Gierens, K., Unterstrasser, S., Jessberger, P., Petzold, A., Voigt, C., and Gayet, J.-F.: Effective Radius of Ice Particles in Cirrus and Contrails, Journal of the Atmospheric Sciences, 68, 300 – 321, https://doi.org/10.1175/2010JAS3562.1, 2011.

Thornberry, T. D., Rollins, A. W., Avery, M. A., Woods, S., Lawson, R. P., Bui, T. V., and Gao, R.-S.: Ice water content-extinction relationships and effective diameter for TTL cirrus derived from in situ measurements during ATTREX 2014, Journal of Geophysical Research: Atmospheres, 122, 4494-4507, https://doi.org/10.1002/2016JD025948, _eprint:https://onlinelibrary.wiley.com/doi/pdf/10.1002/2016JD025948, 2017.

Unterstrasser, S. and Gierens, K.: Numerical simulations of contrail-to-cirrus transition – Part 2: Impact of initial ice crystal number, radiation, stratification, secondary nucleation and layer depth, Atmospheric Chemistry and Physics, 10, 2037–2051, https://doi.org/10.5194/acp-10-2037-2010, publisher: Copernicus GmbH, 2010.

Voigt, C., Schumann, U., Jeßberger, P., Jurkat, T., Petzold, A., Gayet, J.-F., Krämer, M., Thornberry, T., and Fahey, D.: Extinction and optical depth of contrails, Geophysical Research Letters, 38, 1–5, https://doi.org/10.1029/2011GL047189, 2011.

Voigt, C., Kleine, J., Sauer, D., Moore, R. H., Bräuer, T., Le Clercq, P., Kaufmann, S., Scheibe, M., Jurkat-Witschas, T., Aigner, M., Bauder, U., Boose, Y., Borrmann, S., Crosbie, E., Diskin, G. S., DiGangi, J., Hahn, V., Heckl, C., Huber, F., Nowak, J. B., Rapp, M., Rauch, B., Robinson, C., Schripp, T., Shook, M., Winstead, E., Ziemba, L., Schlager, H., and Anderson, B. E.: Cleaner burning aviation fuels can reduce contrail cloudiness, Commun Earth Environ, 2, 1–10, https://doi.org/10.1038/s43247-021-00174-y, number: 1 Publisher: Nature Publishing Group, 2021.

---

## Referee Report (RR1)

Review of revision "Differences in microphysical properties of cirrus at high and mid-latitudes", EGUsphere 2023-374, by Castro, Jurkat-Witschas, Afchine, Grewe, Hahn, Kirschler, Kramer, Lucke, Spelten, Wernli, Zoger, and Voigt.

Overall, I like your responses to my comments. I have a few points below.

The suggestion to use 10 second averages is because low concentrations of small particles may not be included in particle size distribution representations. 0.025 cm-3 (one particle sampled by the CDP is 25/liter. That's typically the total concentration of cirrus ice particles. Perhaps you could do a simple 10 second averaging to see what the effect would be. I do recognize that the path length of this 10 second sample may be 2 km but, that's okay for this exercise.

Liquid origin. If your suggestion that "liquid origin cirrus" is reasonable, you should plot out the aircraft vertical velocities during the penetrations. Do you see updrafts >0.25 m/s approximately that could be used to check your hypothesis about liquid origin cirrus.

Your responses to Darrel Baumgardner's review are good. We'll let him comment on that.

Figure 8. Could you add another panel (d) that shows the relationship between extinction and ice water content for the different combinations.

---

## Author Response (AR2)

We would like to thank the reviewers and editor for studying carefully the responses and improvements we did on the manuscript and for their further comments and suggestions. We include below the comments of Referee #1 in black and our responses in blue. Copied text from the manuscript is marked in italics and underlined sentences correspond to changes in the manuscript.

**Questions and responses to Referee 1**

Review of revision "Differences in microphysical properties of cirrus at high and mid-latitudes", EGUsphere 2023-374, by Castro, Jurkat-Witschas, Afchine, Grewe, Hahn, Kirschler, Kramer, Lucke, Spelten, Wernli, Zoger, and Voigt.

Overall, I like your responses to my comments. I have a few points below.

We thank the reviewer for his positive assessment.

The suggestion to use 10 second averages is because low concentrations of small particles may not be included in particle size distribution representations. 0.025 cm-3 (one particle sampled by the CDP is 25/liter. That's typically the total concentration of cirrus ice particles. Perhaps you could do a simple 10 second averaging to see what the effect would be. I do recognize that the path length of this 10 second sample may be 2 km but, that's okay for this exercise.

Following the reviewer's and editor's suggestion to provide a simple comparison of the three averaging options (1s, 2s and 10s) we looked at two examples to better illustrate the differences and compare the number concentration (N) and effective diameter (ED). The first example (first 2 graphs) shows 10 minutes of a continuous cirrus sequence with some embedded contrail crossings (with enhanced N over 1cm-3 and a reduction in ED). The measurement occurred at constant altitude (around 11 km, gray line) and temperature (around -52 °C, yellow). The blue, orange and green lines represent the N for 1, 2 and 10-sec averages. We observe that both 2-s and 10-s follow the profile of the 1-s line, but the 10-s line often locates the N peak wrongly and sometimes even misses it, while the 2-s reduces slightly the maximum values but reproduces correctly each peak. The 10-s averaging for the ED shows more clearly that the reductions in the ED are not captured.

[Figure]

The next 25 minutes sequence shows the end of a cirrus leg and captures another one at a lower altitude and higher temperature, with also slightly higher N. We also observe that in this case, there are some patches where the cloud is not homogeneous and has very low number concentration with erratic time instants of free air. The 2-s average line, in this case, helps to smooth the scatter in these patches, as well as at the beginning or end of a cloud sequence. In the end, we find that 2-s average does not differ much from the 1-s and therefore makes it comparable to other studies, which usually show cloud properties in 1Hz and, at the same time, gives us the advantage to slightly homogenize certain features, particularly in regions of low concentrations. Depending on the purpose of the study, 10-s average could be an interesting choice. However, in our case, the 10-sec average would homogenize too much and avoid the identification of contrail encounters, which was also an object of our study. We include in the manuscript the following sentences to clarify the choice of 2-s average:

*In general, studies usually use data directly in 1-Hz sample rate. We apply a 2-s mean in order to improve the statistical significance of the low particle concentrations. A larger averaging interval (e.g. 5 or 10 seconds) corresponds to a large horizontal extension (~ 2 km for 10-s averages), where local inhomogeneities can be present, and therefore, it excessively attenuates certain features that are of interest, such as contrails, for example.*

[Figure]

Liquid origin. If your suggestion that "liquid origin cirrus" is reasonable, you should plot out the aircraft vertical velocities during the penetrations. Do you see updrafts >0.25 m/s approximately that could be used to check your hypothesis about liquid origin cirrus.

To determine whether the cirrus we measured were of in situ or liquid origin, we use the definition from Luebke et al., (2016), Krämer et al., (2016) and Wernli et al., (2016). In particular, we applied the method described in Wernli et al., (2016), which uses backward trajectories to verify the presence of liquid or ice water content (LWC/IWC). In the graphs below, we show an example of the cloud water content (CWC) along two backward trajectories, one corresponds to an in-situ origin cirrus (a) and the second one refers to a liquid origin cirrus (b). The red star marks the cloud formation. We classify the measurements associated to trajectory (b) as liquid origin, since LWC content was present at some point since the cloud formation point. Therefore, we do not set here a requirement for the updrafts. For example, in this case, the maximum value of the updraft along the trajectory is ~ 0.09 m/s. We briefly discuss in the text about the updraft velocities regarding the Fig. S3 (originally S2) in the Supplement, where we show frequency distributions of the updrafts along the backward trajectories

and at the measurement points for the four groups (in-situ origin ML cirrus, liquid origin ML cirrus, in situ origin HL cirrus and liquid origin HL cirrus). The maximum values along the trajectories are higher for the liquid origin groups than for the in-situ origin. When considering all values along the trajectories, we also find broader distributions for the liquid origin cases. However, the trend becomes less clear when looking at the instantaneous updraft measurements. We do not find very useful the measurement of the vertical velocities during the penetrations, since they are just as snapshot of the "present" moment and do not provide much information about the cloud history. In an aged origin cloud, it would be a weak indicator of its origin or formation pathway.

[Figure]

Your responses to Darrel Baumgardner's review are good. We'll let him comment on that.

Figure 8. Could you add another panel (d) that shows the relationship between extinction and ice water content for the different combinations.

We thank the reviewer for this suggestion and the following plot shows the relationship between the extinction coefficient and ice water content for the four groups, as defined in Fig. 8 from the text. The linear fits of the four groups matches the ED=200 µm line and the slope of the fits are slightly below 1. A small difference between the in situ and liquid origin cirrus groups can be observed, for the same extinction, the IWC of the liquid origin cirrus is larger, what is in line with previous observations and the results of our study. We see a good agreement with the relationships shown in Heymsfield et al., 2014, particularly with the subplots for the temperature range between -60 and -50 °C and -50 and -40 °C of their Figs. 6 and 7, which is the temperature range of our data. However, the analysis performed in the study of Heymsfield et al., 2014 includes the comparison of various methods, a broad range of temperatures and several campaigns, which is beyond the scope of this study. Since the extinction coefficients are already shown in the panel (c), we would not include the panel in Fig. 8 of the manuscript, if the reviewer sees no urgent need.

[Figure]

**References**

Heymsfield, A., D. Winker, M. Avery, M. Vaughan, G. Diskin, M. Deng, V. Mitev, and R. Matthey, 2014: Relationships between Ice Water Content and Volume Extinction Coefficient from In Situ Observations for Temperatures from 0° to −86°C: Implications for Spaceborne Lidar Retrievals. J. Appl. Meteor. Climatol., 53, 479–505, https://doi.org/10.1175/JAMC-D-13-087.1.

Krämer, M., Rolf, C., Luebke, A., Afchine, A., Spelten, N., Costa, A., Meyer, J., Zöger, M., Smith, J., Herman, R. L., Buchholz, B., Ebert, V., Baumgardner, D., Borrmann, S., Klingebiel, M., and Avallone, L.: A microphysics guide to cirrus clouds – Part 1: Cirrus types, Atmospheric Chemistry and Physics, 16, 3463–3483, https://doi.org/10.5194/acp-16-3463-2016, 2016.

Luebke, A. E., Afchine, A., Costa, A., Grooß, J.-U., Meyer, J., Rolf, C., Spelten, N., Avallone, L. M., Baumgardner, D., and Krämer, M.: 770 The origin of midlatitude ice clouds and the resulting influence on their microphysical properties, Atmospheric Chemistry and Physics, 16, 5793–5809, https://doi.org/10.5194/acp-16-5793-2016, 2016.

Wernli, H., M. Boettcher, H. Joos, A. K. Miltenberger, and P. Spichtinger (2016), A trajectory-based classification of ERA-Interim ice clouds in the region of the North Atlantic storm track, Geophys. Res. Lett., 43, 6657–6664, doi:10.1002/2016GL068922.